

# Resolving phase transitions with discontinuous Galerkin methods

## Eduardo Grossi and Nicolas Wink

Institut für Theoretische Physik, Universität Heidelberg,
Philosophenweg 16, 69120 Heidelberg, Germany

## Abstract

We demonstrate the applicability and advantages of Discontinuous Galerkin (DG) schemes in the context of the Functional Renormalization Group (fRG). We investigate the $O(N)$-model in the large $N$ limit. It is shown that the flow equation for the effective potential can be cast into a conservative form. We discuss results for the Riemann problem, as well as initial conditions leading to a first and second order phase transition. In particular, we unravel the mechanism underlying first order phase transitions, based on the formation of a shock in the derivative of the effective potential.

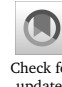
# 1 Introduction

The coherent description of strongly correlated quantum systems is one of the great challenges of modern theoretical physics. While great progress has been achieved in the last decades, theories like the Hubbard model or QCD still provide many challenges to overcome, due to their strongly correlated nature. Different methods usually have different strengths and complement each other. Functional Methods are excellent at uncovering physical mechanisms and relevant degrees of freedom. In particular, the Functional Renormalization Group (fRG), introduced in [1–3], provides a very powerful tool to investigate the phase structure in strongly correlated theories, ranging from condensed matter systems to quantum gravity. Truncations of the underlying functional partial differential equation within this framework usually result in system of convection-diffusion equations. Despite their successful investigation in a tremendous amount of theories, their numerical treatment with non-analytic solutions has so far not been studied in detail. In turns out that this situation is relevant in the vicinity of a first order phase transition, which demands the usage of suitable numerical tools. The leading order equations within such truncations governing the Renormalization Group (RG) evolution can be cast into a conservative form, which is very similar, in some aspects, to the equations studied in hydrodynamics and in general, mathematical physics. This already suggests the use of suitable numerical techniques, incorporating e.g. the directed flow of information.

Equations of this type generally lead to the formation of a discontinuity in the solution. Therefore, the applied numerical scheme has to be able to handle non-analyticities appropriately. A standard and robust choice is the Finite Volume (FV) scheme, where the equations are solved in a collection of small volumes. Schemes of this type are capable of treating discontinuities in a stable manner. However, they are lacking in accuracy, since it is challenging to adopt higher order formulations while preserving numerical stability. On the other hand, Pseudo-Spectral methods are designed to have an arbitrarily high order accuracy, since the solution is obtained in a functional space spanned by a suitable basis. However, non-analyticities in the solution usually lead to spurious oscillations, which may ruin the stability of the scheme. Discontinuous Galerkin (DG), introduced in [4–8], schemes utilize the strengths of both methods. The domain is decomposed in small volumes; therefore, discontinuities are well treated, and the solution is approximated locally within an appropriate basis to achieve high accuracy. The demand for high accuracy in fRG calculations, combined with the convection dominated nature of the equations, makes DG schemes a natural choice.

Here, we present the application of Discontinuous Galerkin methods to the $O(N)$-model for $N \gg 1$, within the fRG framework. The paper is organized as follows: To close the gap between the fRG as well as the DG community, and languages therein, we introduce both fields in Section 1.1 and Section 1.2, respectively. In Section 2 the $O(N)$-model in the large

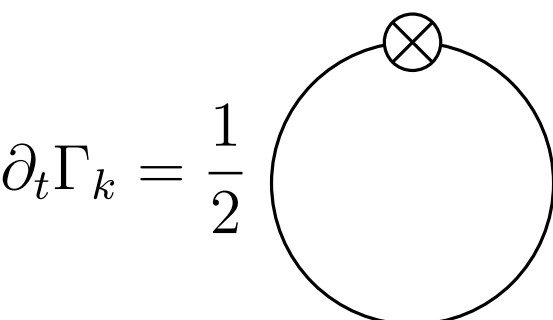

Figure 1: Graphical representation of the Wetterich equation (1). The line represents the full, regularized propagator $G_k$, while the crossed circle denotes the regulator derivative insertion $\partial_t R_k$, with respect to the RG-time $t = -\ln\left(\frac{k}{\Lambda}\right)$.

$N$ limit and the applications of DG methods to the flow equation of the effective potential are discussed. Numerical results are presented in Section 3, starting with the associated Riemann problem in Section 3.1. Initial conditions quartic in the field, leading to a second order phase transition, are presented and compared to results from the method of characteristics in Section 3.2. Increasing the order of the potential in the initial conditions allows for a first order phase transition, which is discussed in Section 3.3. Finally, we close with conclusions and future perspectives in Section 4.

## 1.1 The functional renormalization group

Here we give a very brief introduction to the fRG, which should be sufficient to outline the underlying basic ideas, more complete introductions/reviews can be found in e.g. [9–11].

The fRG implements the idea of Wilsonian renormalization, while providing a suitable regularization of the underlying Quantum Field Theory (QFT). This results in an exact equation (1) for the effective average action $\Gamma_k[\Phi]$, where $\Phi$ is a vector collecting all quantum fields of the theory under investigation. The effective average action $\Gamma_k[\Phi]$ is the generator of One-Particle Irreducible (1PI) correlation functions, where all fluctuations up to momentum scale $p^2 \sim k^2$ have been taken into account. In the fRG momentum shells are integrated out around the momentum scale $k$ and is described by [1],

$$\partial_k \Gamma_k[\Phi] = \frac{1}{2}\mathrm{Tr}\left\{\left(\frac{1}{\Gamma_k^{(2)}[\Phi] + R_k}\right)_{ij}(\partial_k R_k)_{ij}\right\}, \tag{1}$$

where the trace integrates over momenta and internal spaces, such as color space for a gauge theory. $R_k$ is the regulator that acts as like a mass and therefore regularizes the effective propagator $G_k = \left(\Gamma_k^{(2)}[\Phi] + R_k\right)_{ij}^{-1}$. The indices describe the different field and have to be summed over. Finally, the scale derivative of the regulator $\partial_k R_k$ acts as a UV-cutoff and renders the theory UV finite.

In this manner, (1) interpolates between the classical action $\Gamma_k[\Phi] \to S[\Phi]$ for $k \to \infty$ and the full quantum effective action $\Gamma_k[\Phi] \to \Gamma[\Phi]$ for $k \to 0$, which is the generating functional for all correlation functions of the quantum theory and therefore its solution. It is convenient to work with a dimensionless RG-time

$$t = -\ln\left(\frac{k}{\Lambda}\right), \tag{2}$$

which also captures the natural scaling of dimensionless couplings. In (2) we have included an additional minus sign compared to most fRG related works, to have a positive RG-time

evolution. The reference scale $\Lambda$ is also usually used to as initial scale for (1), where one assumes that the classical theory describes the underling theory sufficiently well, for more details on this and the related issue of RG consistency, see e.g. [12]. Equation (1), and related flow equations, are often depicted graphically, the representation of (1) is shown in Figure 1.

Finding suitable truncations, i.e. an ansatz for the effective average action $\Gamma_k[\Phi]$, is not an easy task and usually one has to follow physical intuition, taking correlations functions of the relevant degrees of freedom into account. This corresponds to an expansion of correlation functions in field space, as well as momentum space. Truncations that keep dominantly the field dependence, while taking the momentum dependence only to a low order into account, are usually referred to as *derivative expansion*, see e.g. [9, 13–32]. On the other hand, truncations that dominantly resolve the momentum dependence of correlation functions, but quite often also resolve field dependencies, are usually referred to as *vertex expansion*, see e.g. [33–47]. In practice, often a mixed approach must be used to achieve quantitative results, while the qualitative features of the system under investigation can often be captured by rather simple truncations.

In practice, the partial different equation part of the resulting equations for a given ansatz are usually non-linear convection-diffusion equations. During the most part of the flow, these equations are also convection dominated, since (1) is already designed to be dominated by a single scale, set by the RG-time $t$, in all quantities. We will come back to this point in Section 3.2.2. Additionally, in our application to the $O(N)$-model in the large $N$ limit, this becomes exact, i.e. it the resulting flow equation is a convection equation, c.f. (6). Moreover, it can be cast into a conservative form, therefore we will restrict the introduction to DG methods in some parts to conservation laws to keep it simple. Having the equation in a conservative form is particularly convenient, since it allows us to understand how a jump discontinuity in the solution forms and propagates.

## 1.2 Discontinuous Galerkin methods

We review some basic facts about DG schemes following [48], for simplicity we restrict ourselves to one spatial dimension. For an introduction to foundations of numerical methods for PDEs, that DG schemes build upon, e.g. Finite Element and Finite Volume Methods, the reader is referred to [49–51].

The problem is considered over a domain $\Omega$, with boundary $\partial\Omega$, approximated by a computational domain $\Omega_h$, composed by $K$ non-overlapping elements $D^k$

$$\Omega \simeq \Omega_h = \bigcup_{k=1}^{K} D^k. \tag{3}$$

The approximate solution is then represented by

$$u(t,x) \simeq u_h(t,x) = \bigoplus_{k=1}^{K} u_h^k(t,x). \tag{4}$$

The local solution is approximated in each element by a polynomial of degree $N$

$$u_h^k(t,x) = \sum_{n=1}^{N+1} \hat{u}_n^k(t)\psi_n(x) = \sum_{i=1}^{N+1} u_h^k(t,x_i^k)l_i^k(x), \tag{5}$$

where the first version is the modal expansion, expressing the solution in terms of a local polynomial basis $\psi_n(x)$. The second approximation in (5) is referred to as nodal expansion, which introduces $N+1$ grid points $x_i^k$ and $l_i^k(x)$ is the associated Lagrange polynomial. For

calculations, we use the usual Legendre basis in the modal expansions and the Legendre-Gauss-Lobatto quadrature points as grid points in the nodal expansion. A few more details are given in Appendix A.

The main task at hand is to solve the conservation law, which turns out to be the relevant form of the equation in our application, posed as an initial value problem

$$\partial_t u + \partial_x f(u) = 0, \tag{6}$$

where we assume the flux $f$ to be convex, and it may also depend on the time $t$. We now require that the residual is orthogonal to the basis function *locally* in each element

$$\int_{D^k} \left( \partial_t u_h^k + \partial_x f_h^k(u_h^k) \right) \psi_n \, \mathrm{d}x = 0, \tag{7}$$

i.e. the space of test functions for which we require the orthogonality of the residual of the equation is chosen to be the same as the function space of the solution approximation. Choosing the test function space and the function space of the solution equal is referred to Galerkin method, hence the name Discontinuous Galerkin methods. Additionally, due to the disconnected nature of the approach, (7) has still more degrees of freedom than equations. To resolve this, we integrate (7) by parts and obtain the locally defined weak formulation

$$\int_{D^k} \left( \left( \partial_t u_h^k \right) \psi_n - f_h^k(u_h^k) \partial_x \psi_n \right) \mathrm{d}x = - \int_{\partial D^k} f^* \cdot \hat{\mathbf{n}} \, \psi_n \, \mathrm{d}x, \tag{8}$$

where we have already replaced the flux on the right-hand side with an approximation thereof, the numerical flux $f^*$, discussed below. In the one dimensional case the element boundary $\partial D^k$ consists of two points and the outward pointing normal vector is $\hat{\mathbf{n}} = \pm 1$. Integrating (8) once more by parts we obtain the strong formulation

$$\int_{D^k} \left( \partial_t u_h^k + \partial_x f_h^k(u_h^k) \right) \psi_n \, \mathrm{d}x = \int_{\partial D^k} \hat{\mathbf{n}} \cdot \left( f_h^k(u_h^k) - f^* \right) \psi_n \, \mathrm{d}x, \tag{9}$$

which we also use throughout this work for all numerical calculations. It is important to stress that the solution is only defined element wise. The value of the flux at the boundary is not necessarily equal to the value of the flux on a boundary node, but depends on the solution of all elements sharing that particular intersection, i.e. two in one dimension. Therefore, the numerical fluxes are define on each intersection and depend non-trivially on the value of the approximate solution on all adjacent elements. Specifying the numerical flux closes the set of equations. For the case of a scalar conservation law one can rely on the results for the choice of numerical fluxes obtained in Finite Volume Methods, where the subject has been studied extensively, see e.g. [48, 50]. The main requirements are consistency, i.e. $f^*(u, u) = f(u)$, and monotonicity. We will refrain here from a more detailed discussion on numerical fluxes and rather state that we work with the Local Lax-Friedrichs flux [52] given by

$$f^*(u_h^-, u_h^+) = \{\{f_h(u_h)\}\} + \frac{C}{2} [[u_h]], \tag{10}$$

where an index $-$ denotes the interior information of the element while an index $+$ denotes the exterior information of the element. The brackets denote the average and jump, respectively

$$\begin{aligned} \{\{u\}\} &= \frac{1}{2} \left( u^- + u^+ \right), \\ [[u]] &= \hat{\mathbf{n}}^- u^- + \hat{\mathbf{n}}^+ u^+. \end{aligned} \tag{11}$$

The constant $C$ in (10) is chosen as the maximal wave speed in a local manner as

$$C \geq \max_{u^{\pm}} |\partial_u f(u)| \,, \tag{12}$$

which is related to the fastest propagating mode. To be more precise, the numerical flux also ensures the convergence to the correct result in situations where discontinuities are present, i.e. it ensures the convergence to the correct solution. This solution can be interpreted as being fixed by an entropy condition or as the inviscid limit of the equation with an infinitesimal viscosity term.

Additionally, boundary conditions have to be specified for all inflow boundaries, given by $\hat{\mathbf{n}} \cdot (\partial_u f) < 0$.

Finally, we would like to note that (9) can be written as

$$\mathcal{M}^k \partial_t u_h^k + \mathcal{S}^k f_h^k = \left[ l^k(x)(f_h^k - f^*) \right]_{x_l^k}^{x_r^k}, \tag{13}$$

resulting in a fully discretized scheme, where we have introduced the two matrices

$$\mathcal{M}_{ij}^k = \int_{D^k} l_i^k(x) l_j^k(x) \, dx \,,$$

$$\mathcal{S}_{ij}^k = \int_{D^k} l_i^k(x) \partial_x l_j^k(x) \, dx \,. \tag{14}$$

In the usual manner, the discretized operators (14) are calculated for a reference element and the appropriate mappings to the actual elements invoked.

## 2 $O(N)$-model

We consider the $O(N)$ model in $d$-dimensional Euclidean spacetime. The field can be described by a collection of $N$ scalar fields $\phi_a(x)$ with $a = 1, \cdots, N$. The action for $N$ scalar field is

$$S = \int d^d x \left\{ \frac{1}{2}(\partial_\mu \phi_a)^2 + V(\rho) \right\} , \tag{15}$$

where $V(\rho)$ is the interacting potential. The $O(N)$ symmetry acts on the fields as an orthogonal transformation $\phi_a \to O_{ab}\phi_b$. Consequently, the $O(N)$-invariant terms are those constructed by the modulus of the fields $\phi_a\phi_a$. Given this symmetry, the potential is restricted to depend only on $O(N)$-invariant terms, namely the combination $\rho = \frac{1}{2}\phi_a\phi^a$. This quite simple model can nevertheless describe an immense variety of physical system at different energy scale, from the Higgs sector of the standard model to the phase transition in ferromagnets. The O(N) model is the prototype used to investigate phase transition in different type of systems. For $N = 4$ and $d = 4$ the model describes the scalar sector of the standard model (at zero Yukawa couplings). It also captures the essential features of the chiral phase transition in QCD in the limit of two flavors. Moving to lower energy scales, $N = 3$ corresponds to the Heisenberg model that describes the phase transition of a ferromagnet. In condensed matter, i.e. $d \leq 3$, there are many other applications of the O(N) model, as for example $N = 2$ can describe the helium superfluid phase transition or $N = 1$ is the liquid-vapour transition. The motivation for the wide range of applicability of such a simple model comes from the universal behavior of physical systems close to a phase transition; under these circumstances the microscopic details of a system are not important, only a few characteristics like the underlying symmetry govern the physics close to the phase transition.

For this reason, the O(N) model is the perfect prototype to understand relevant mechanisms that govern a phase transition.

## 2.1 Flow equations

To derive flow equations, we need to truncate the effective action, i.e. we need to choose an ansatz. Here we work in a *derivative expansion*, i.e. we expand the action in terms of gradients of the field. The zeroth order of the expansion is usually referred to as *Local potential approximation* (LPA). For our intended purpose, i.e. $N \gg 1$, this approximation becomes exact [53]. Within LPA the ansatz for the effective action is given by

$$\Gamma_k[\phi] = \int_x \left\{ \frac{1}{2}(\partial_\mu \phi_a)^2 + V(t,\rho) \right\}, \tag{16}$$

where $V(t,\rho)$ is the effective potential, which depends only on the RG-time as well as the $O(N)$ invariant $\rho = \frac{1}{2}\phi_a \phi^a$. Having specified the ansatz for the effective action, we can derive a PDE for the effective potential by evaluating the right-hand side of (1). This requires the knowledge of the regularized propagator, or equivalently the two-point function

$$\Gamma_{ab}^{(2)}(t,\rho,p) = \left[ p^2 + V'(t,\rho) \right] \delta_{ab} + 2\rho V''(t,\rho) \delta_{aN} \delta_{bN}, \tag{17}$$

where we introduced the shorthand notation $V'(t,\rho) = \partial_\rho V(t,\rho)$ and specified the field direction where it can acquire a non-vanishing expectation value. Plugging (17) into (1), using a regulator that's diagonal in field space, and carrying out the trace up to the momentum integral one obtains

$$\partial_t V(t,\rho) = \frac{1}{2} \int_q \left[ \left( \frac{N-1}{q^2 + V'(t,\rho) + R_k(q)} + \frac{1}{q^2 + V'(t,\rho) + 2\rho V''(t,\rho) + R_k(q)} \right) \partial_t R_k(q) \right]. \tag{18}$$

As regulator we chose the usual Litim regulator

$$R_k(p) = (k^2 - p^2)\Theta(k^2 - p^2), \tag{19}$$

which provides the optimal [54] choice in LPA. Additionally, we rescale $\rho$ and $V(t,\rho)$ with factors of $1/N-1$,

$$\rho \rightarrow (N-1)\rho, \tag{20}$$
$$V(t,\rho) \rightarrow (N-1)V(t,\rho),$$

which allows for easy access to the large $N$ limit. Putting (18), (19) and (20) together, the integration becomes trivial and we arrive at the flow equation for the effective potential

$$\partial_t V(t,\rho) = -A_d(\Lambda e^{-t})^{d+2} \left( \frac{1}{(\Lambda e^{-t})^2 + V'(t,\rho)} + \frac{1}{N-1} \frac{1}{(\Lambda e^{-t})^2 + V'(t,\rho) + 2\rho V''(t,\rho)} \right), \tag{21}$$

with $A_d = \Omega_d (2\pi)^{-d}/d$ and $\Omega_d = 2\pi^{d/2}\Gamma(\frac{d}{2})^{-1}$ is the volume of a $d-1$ dimensional sphere. Please note that $\Gamma$ denotes only in this context the usual Gamma function. Due to the rescaling (20) it is very easy to go to the limit $N \gg 1$, i.e. we drop the last term in (21)

$$\partial_t V(t,\rho) = -A_d \frac{(\Lambda e^{-t})^{d+2}}{(\Lambda e^{-t})^2 + V'(t,\rho)}. \tag{22}$$

Before doing calculations we still have to fix the dimension $d$ as well as the initial UV-scale $\Lambda$ in (22). For the dimension we chose $d = 3$, enabling us to investigate phase transitions of first and second order. The choice of the UV-cutoff is completely arbitrary, and therefore we chose $\Lambda = 1$ a.u.. Where a.u. denotes arbitrary unit, and consequently all dimensionful quantities are rescaled by appropriate powers of $\Lambda$ to make them dimensionless in a practical manner, but not from an RG perspective. To keep the notation concise, the rescaled quantities are not denoted in a different manner but are understood dimensionless for the remainder of this work. This theory has been studied extensively within the fRG, see e.g. [55–57].

## 2.2 Numerical treatment

The flow equation (22) is non-linear, since the derivative of the potential with respect to the field expectation value appears on the right-hand side in a non-linear manner. However, for numerical purposes, and to apply DG schemes, it is preferable to formulate the problem in conservative form. As $V(t,\rho)$ is related to the zero-point energy of the underlying QFT, its equation should not depend on itself, as it is already the case in (1), and consequently also in (22). As a direct consequence, the first derivative of the potential is a conserved quantity, in the sense of (6). Therefore, we introduce the derivative of the potential as a new variable

$$u(t,\rho) = \partial_\rho V(t,\rho),\tag{23}$$

as well as the flux

$$f(t,u) = A_d \frac{(\Lambda e^{-t})^{d+2}}{(\Lambda e^{-t})^2 + u}.\tag{24}$$

Because all derivatives of a solution of a PDE must also satisfy the PDE, we can take a derivative of (22) to obtain an equation for the derivative of the potential $u$, which is easily expressed in terms of the flux (24)

$$\partial_t u + \partial_\rho f(t,u) = 0.\tag{25}$$

Therefore, we are left with the task of solving a scalar conservation law with a flux that depends explicitly on the RG-time, allowing us to make immediate use of the spatial discretization presented in Section 1.2. As boundary condition we need to specify the flux at large field values, the inflow boundary. However, in this case it is naturally suppressed for physical initial conditions, c.f. (24). Therefore, we have fixed the flux at the boundary to a flux with the initial derivative of the potential. Additionally, we have verified explicitly that we obtain numerically equivalent results by setting the flux to zero at the boundary. Both cases are therefore sufficiently close to the true boundary conditions, i.e. fixing the flux to the initial conditions at infinite field values. It is noteworthy that the flux (24) is convex for all RG-times. Additionally, we would like to note that the weak formulation has already been used in the context of the fRG in [58].

The time dependence is treated with the method of lines, i.e. we use the usual machinery of ordinary differential equations (ODE). Preferably one uses a suitable explicit scheme in this context as numerical stability can be ensured, when the size of the time steps respects the associated Courant-Friedrichs-Lewy (CFL) condition, see e.g. [48]. The condition states that stability is ensured as long as the physical light cone of the system is contained in the numerical one, see e.g. [51]. Therefore, it is related to the propagation of information and is bounded by the physics of the system, e.g. in relativity it should always be less than the speed of light. However, the equation encountered here is quite peculiar from this point of view, since the characteristic speed of information $\partial_u f$ is not bounded and time dependent. As we will show in section Section 3.2.2 in the limit $t \to \infty$ the wave speed generally diverges exponentially fast for a subdomain. Therefore, the time step required by the CFL condition also decreases exponentially fast and becomes infeasible in this region. This can be circumvented using implicit methods, and we resorted to the family of Backward Differentiation Formula (BDF) methods, where we used SUNDIALS [59] through its Mathematica interface [60]. Additionally, we have compared our results for all qualitatively different solutions to a strong stability persevering scheme Runge-Kutta scheme [61], with a time step chosen through the CFL condition.

The numerical schemes outlined here are generally applicable, in particular also to future applications in relativistic hydrodynamics [51, 62–66]. Additionally, the high-performance aspects of DG methods, see e.g. [67–69], are a promising perspective for complex fRG settings, where the computational complexity grows fast.

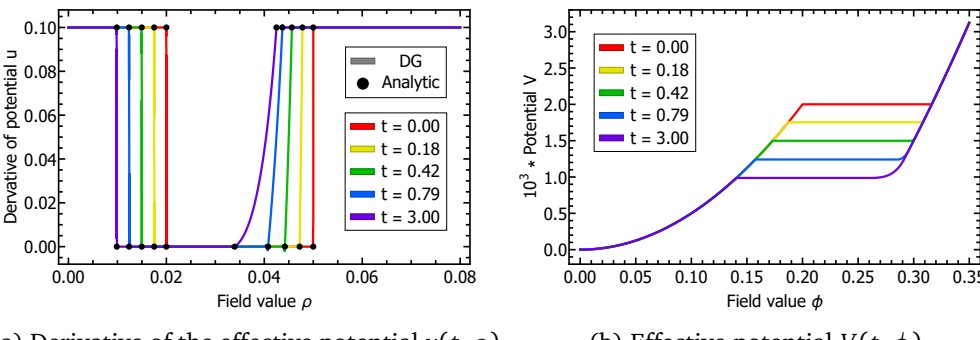

(a) Derivative of the effective potential $u(t,\rho)$     (b) Effective potential $V(t,\phi)$

Figure 2: RG-time evolution of $u(t,\rho)$ for the Riemann problem (32). The dots represent the analytic result of the position of the shock (28) and the boundaries of the rarefaction wave (31). The numerical results were obtained with $K = 1500$ elements and a local accuracy of order $N = 3$. The results for the derivative of the potential were post-processed with a minmod limiter.

## 3  Results

### 3.1  Riemann problem

The Riemann problem is a well-known problem, usually studied in fluid dynamics, and is designed to understand how discontinuities arises and evolve. It consists of finding the solution to the PDE at hand with piecewise constant initial condition

$$u(0,\rho) = \begin{cases} u_{\mathrm{L}}, & \rho \leq \xi_0, \\ u_{\mathrm{R}}, & \rho > \xi_0, \end{cases} \tag{26}$$

where we restrict ourselves to the case $u_{\mathrm{L/R}} \geq 0$, due to the possibly divergent flux (24) otherwise. For these initial conditions, the solution will either develop a shock or a rarefaction wave, depending on whether the characteristic curves intersect or not, respectively. For our problem at hand, i.e. equation (25) together with the flux (24), information is propagating from large $\rho$ to small $\rho$, therefore we will have a propagating shock when $u_{\mathrm{L}} > u_{\mathrm{R}}$, and consequently a rarefaction wave when $u_{\mathrm{L}} < u_{\mathrm{R}}$.

#### 3.1.1  Analytic investigation

For the case of a propagating shock the position $\xi$ of it must satisfy the Rankine-Hugoniot condition, see Appendix C for a derivation or e.g. [48],

$$\frac{\mathrm{d}\xi(t)}{\mathrm{d}t} = \frac{[\![f]\!]}{[\![u]\!]}, \tag{27}$$

where the difference bracket is defined in (11). Since the flux (24) does not depend on field space, the solution will trivially stay piecewise constant. From (27) it can immediately be seen that the speed of shock is time dependent and exponentially suppressed for large times, since it is the case for the flux (24), independent of the values of $u_{L/R}$. The differential equation (27) can be solved analytically, where we employ as initial condition $\xi(t=0) = \xi_0$. The solution of (27) together with (24) in $d$ dimensions is

$$\xi(t) = \xi_0 + A_d \frac{\Lambda^d}{d(u_{\mathrm{R}} - u_{\mathrm{L}})} \left[ {}_2\tilde{F}_1\left(\frac{u_{\mathrm{R}}}{\Lambda^2}\right) - {}_2\tilde{F}_1\left(\frac{u_{\mathrm{L}}}{\Lambda^2}\right) - e^{-dt}\,{}_2\tilde{F}_1\left(\frac{u_{\mathrm{R}}}{\Lambda^2}e^{2t}\right) + e^{-dt}\,{}_2\tilde{F}_1\left(\frac{u_{\mathrm{L}}}{\Lambda^2}e^{2t}\right) \right], \tag{28}$$

where $_2\tilde{F}_1(z) = {_2F_1}(1, -\frac{d}{2}, 1 - \frac{d}{2}, -z)$ and $_2F_1$ is the Gaussian or ordinary hypergeometric function. Specifying to $d = 3$, $\Lambda = 1$ and $u_{\mathrm{R}} = 0$ it is possible to simplify (28) considerably

$$\xi(t) = \xi_0 + \frac{1}{6\pi^2}\left[e^{-t} - 1 + \sqrt{u_{\mathrm{L}}}\Big(\cot^{-1}(\sqrt{u_{\mathrm{L}}}) - \cot^{-1}(e^t\sqrt{u_{\mathrm{L}}})\Big)\right]. \tag{29}$$

We have chosen the specific value of $u_{\mathrm{R}} = 0$, because it will be the situation encountered later in the case of a first order phase transition, c.f. Section 3.3. The form (29) gives us access to the infinite RG-time limit

$$\xi_\infty \equiv \xi(t \to \infty) = \xi_0 + \frac{\sqrt{u_{\mathrm{L}}}\cot^{-1}(\sqrt{u_{\mathrm{L}}}) - 1}{6\pi^2}. \tag{30}$$

Therefore, the shock freezes in at large RG-time, and it does so exponentially fast at late times. Where the latter statement can be seen from the expansion of the $\cot^{-1}$ term in (29).

Having discussed the analytic case of a shock wave, we turn now to the case of a rarefaction wave, i.e. $u_{\mathrm{R}} > u_{\mathrm{L}}$. From the perspective of the characteristic curves, the one at the left boundary is moving faster than the one on the right boundary. Compared to the previous case, here the characteristics aren't overlapping, but rather there is a lack of characteristics. Nevertheless, the problem admits a unique solution, but unfortunately due to the explicit time dependence of the flux (24), the explicit solution cannot be constructed in the usual manner. The speed of the boundary points $\xi^{\mathrm{B}}$ however is directly related to the associated characteristics and therefore their explicit solution is easily constructed

$$\begin{aligned}
\xi_{\mathrm{L/R}}^{\mathrm{B}}(t) = \xi_0^{\mathrm{B}} - \frac{A_d\Lambda^d}{2}\Bigg[&\frac{e^{-dt}}{u_{\mathrm{L/R}} + (\Lambda e^{-t})^2} - \frac{1}{u_{\mathrm{L/R}} + \Lambda^2}\\
&- \frac{d(\Lambda e^{-t})^{d-2}}{\Lambda^d(d-2)}{_2F_1}\left(1, 1 - \frac{d}{2}, 2 - \frac{d}{2}, -\frac{u_{\mathrm{L/R}}}{(\Lambda e^{-t})^2}\right)\\
&+ \frac{d\Lambda^{-2}}{d-2}{_2F_1}\left(1, 1 - \frac{d}{2}, 2 - \frac{d}{2}, -\frac{u_{\mathrm{L/R}}}{\Lambda^2}\right)\Bigg].
\end{aligned} \tag{31}$$

Similarly, as for the case of a propagating discontinuity (28), the propagation of the boundaries of the rarefaction wave is also exponentially suppressed and only propagates a finite amount in field space. This can easily be inferred from (31) for $d = 3$. Since we did not encounter any rarefaction waves during our investigation of first and second order phase transition, c.f. Section 3.3 and Section 3.2, we will refrain from an in-depth discussion at this point.

### 3.1.2 Numerical investigation

Having discussed the solution of the Riemann problem at length from an analytic point of view in Section 3.1.1, we now turn to its numerical investigation. Here it is convenient to investigate both cases at the same time. To be more precise, we chose as initial conditions

$$u(0, \rho) = \begin{cases} 0.1, & 0 < \rho < 0.02, \\ 0, & 0.02 < \rho < 0.05, \\ 0.1, & \rho > 0.05. \end{cases} \tag{32}$$

The solution to the initial conditions (32) will evolve a shock in the solution from the jump at $\rho = 0.02$ and a rarefaction wave for the jump at $\rho = 0.05$. Our results are shown in Figure 2. The derivative of the potential $u(t, \rho)$, which is resolved numerically, is shown in Figure 2a

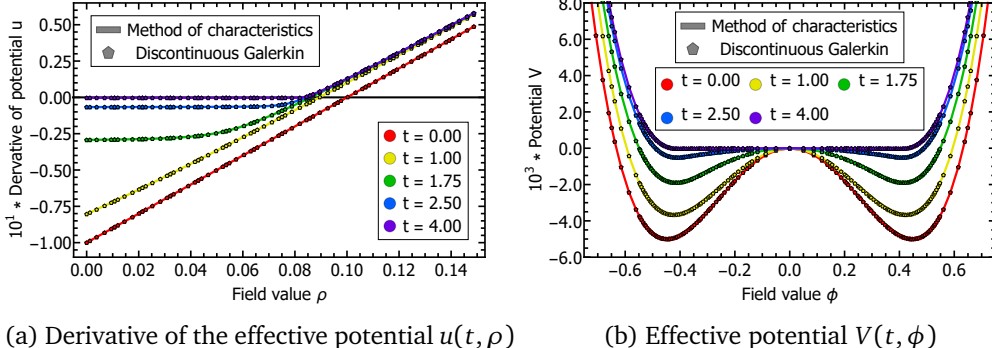

(a) Derivative of the effective potential $u(t, \rho)$   (b) Effective potential $V(t, \phi)$

Figure 3: RG-time evolution for the second order problem (33) for $\lambda_2 = -0.1$. The full lines are a semi-analytic result obtained by the method of characteristics, while the hexagon dots are the respective results obtained by a numerical simulation with $K = 30$ elements and a local accuracy of order $N = 5$.

and the corresponding effective potential, obtained by integrating, in Figure 2b. We find the expected agreement with the analytic results of Section 3.1.1. For the calculation we used $K = 1500$ equally sized elements in the domain $0 \leq \rho \leq 0.08$ with a local interpolation order of $N = 3$ and evolved up to the RG-time $t = 3$. This upper RG-time is already relatively close to the infinite RG-time limit, i.e. the position of the shock as well as the rarefaction wave are effectively frozen in. During the RG-time evolution inevitably spurious Gibbs oscillations will form. Here we simply chose to keep them at a minimal level by using a considerable amount of degrees of freedom and post-process the results with a simple minmod limiter, see e.g. [48], but remnants of the oscillations can still be seen in Figure 2a. Nevertheless, it should be noted that the result still maintains it spectral accuracy, see e.g. [70], i.e. point wise convergence can be recovered from the numerical result. This is done partially by integrating the result within our polynomial basis, which removes all oscillations, c.f. Figure 2b, which is obtained from the result without a limiter. However, for future application we will consider the use of a limiter or utilize a shock capturing scheme, since the numerical approximation of the derivative of the potential $u(t, \rho)$ must become positive semidefinite in the large RG-time limit to avoid potential problems due to an artificially divergent flux.

## 3.2 Second order phase transition

We now turn to the initial conditions usually considered in the context of the $O(N)$-model, i.e. a quadratic potential in the classical action

$$V_{\Lambda}(\rho) = \lambda_2 \rho + \frac{\lambda_4}{2} \rho^2. \tag{33}$$

This is the case usually studied in the literature, and it is well known that the classical action (33) leads to a second order phase transition as a function of $\lambda_2$ for a given positive $\lambda_4$. As our main interest is the investigation of a second order phase transition, we restrict ourselves here to $\lambda_4 = 1$ for the remainder of the section. Additionally, we could always rescale the fields to have $\lambda_4 = 1$ in this case, since (33) has only two free parameters. Utilizing the method of characteristics, for details see Appendix B, it is easy to see that local minima are shifted during the flow

$$\rho_{\min}(t) = \rho_{\min}(0) - \Lambda^{(d-2)} \frac{A_d}{d-2} \left( 1 - e^{(d-2)t} \right), \tag{34}$$

which is independent of the initial conditions. Combining (34) with the initial potential (33), the flow of the effective potential inherits a second order phase transition.

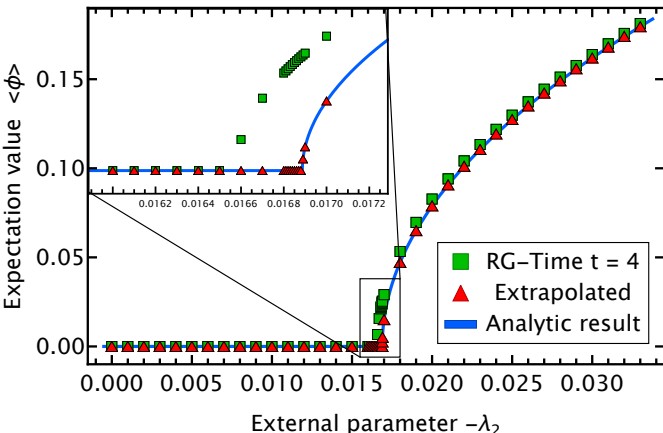

Figure 4: Second order phase transition for the initial conditions (33). The result of the numerical simulation is shown with green squares, the extrapolated result with red triangles and the analytic result by a blue line. A detailed description can be found in the main text.

The RG-time evolution of the effective potential for the case of a finite expectation value, with initial value $\lambda_2 = -0.1$, is shown in Figure 3b. To illustrate the behavior of the individual nodes during the RG-time evolution, we have used only a moderate number of elements, i.e. $K = 30$, and a local approximation order of $N = 5$. However, the elements are not equally sized, but here we already utilize one of the strengths of the DG approach and half of the elements are equally distributed in $0 \leq \rho \leq 0.15$ and the other elements are equally distributed in $0.15 \leq \rho \leq 1$. This distribution of elements ensures that the outer boundary is at sufficiently large field values and our boundary conditions are satisfied, as discussed in Section 2.2, at very little cost. Please note that in Figure 3b the potential is shown as a function of the expectation value of the field $\phi = \sqrt{2\rho}$. Correspondingly, the derivative of the potential for the same calculation is shown in Figure 3a. The maximal RG-time was chosen to be $t = 4$, where all qualitative features have emerged, and only minor quantitative changes occur towards the asymptotic limit $t \to \infty$. The full effective potential has to be convex, which translate to a positive definite derivative of the potential $u \geq 0$ in the infinite time limit. This translates to a flat potential in between the minima, see Figure 3b. How this is realized in the current equation under investigation has been discussed at length in the literature, see e.g. [71–73]. Nevertheless, the numerical stability in the flat region of the potential is a noteworthy advantage of the DG approach.

### 3.2.1 Phase structure

We are now in the position to investigate the phase structure of the theory with classical action (33), where we set $\lambda_4 = 1$, as previously discussed. For all calculations we used $K = 120$ elements and a local interpolation of order $N = 5$. As in the previous case, the elements are not equally distributed. We used 5 equally spaced elements in the interval $0 \leq \rho \leq 0.001$, 15 equally spaced elements in the interval $0.001 \leq \rho \leq 0.01$, 50 equally spaced elements in the interval $0.01 \leq \rho \leq 0.15$ and 50 equally spaced elements in the interval $0.15 \leq \rho \leq 1$. This ensures a good resolution for small field values, and therefore the relevant region in field space at the second order phase transition. The solution is computed up to the RG-time $t = 4$, however, there is no restriction to continue the numerical simulation to larger RG-times. The result of the simulations is shown in Figure 4 with green squares. The final RG-time was also restricted to demonstrate the easy extrapolability of the result to its asymptotic solution

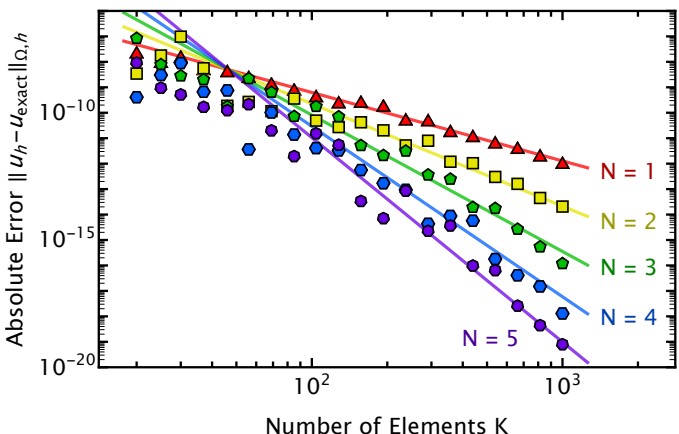

Figure 5: Error of the solution with initial condition (33) ($\lambda_2 = -0.1$) at RG-time $t = 4$ computed in the interval $0 \leq \rho \leq 1$ for different number $K$ of equally sized elements and local approximation order $N$. The symbols show the result of the numerical simulations, while the lines show a $\chi^2$-fit with respect to (39) with the parameters given in Table 2.

at infinite RG-time. For a dimensionful coupling one expects asymptotically an exponential decay

$$\rho_{\min}(t) = \rho_{\min}^{\text{final}} + b\, e^{-ct}, \quad \text{for} \quad t \gg 0. \tag{35}$$

We found compatibility of our numerical results for the position of the minima with (35), which is not very surprising as the analytic solution is given in this form (34). Nevertheless, the form of (35) is a generic feature and valid for couplings with a non-trivial RG-time evolution in this theory, this feature will become relevant in Section 3.3. We have extrapolated the global minimum for each coupling with eleven equally spaced points in the RG-time interval $3 \leq t \leq 4$ according to (35), the result is shown in Figure 4 with blue triangles.

It is very well known that all observables show a power law behavior in the vicinity of a second order phase transition due to the divergent correlation length at the phase transition. This can be parametrized as

$$\langle \phi \rangle = \begin{cases} \alpha \left| \lambda_2 - \lambda_2^{\text{crit}} \right|^{\nu}, & \lambda_2 \leq \lambda_2^{\text{crit}}, \\ 0, & \lambda_2 > \lambda_2^{\text{crit}}, \end{cases} \tag{36}$$

where $\alpha$ is some prefactor, $\nu$ is the critical exponent and $\lambda_2^{\text{crit}}$ is the critical coupling. The exact coefficients can be easily obtained analytically and are given in Table 1, as well as being depicted by a blue line in Figure 4.

Additionally, we have extracted the parameters from our results, extrapolated to infinite RG-time, using a $\chi^2$ minimization. The resulting parameters, including their $1\sigma$ confidence interval, given in terms of the last two digits, are also shown in Table 1. Despite not aiming for a quantitative resolution of the critical area around the phase transition, we obtain an accurate estimate for all parameters. In particular, the error of the critical value of the coupling is only $1.5 \cdot 10^{-7}$, but it should be noted that the critical properties, i.e. the critical exponent in this case, can also be obtained from the fixed point equations, where higher accuracy is easier achievable, see e.g. [27].

### 3.2.2 Propagation of information and approach to convexity

It is instructive to have a closer look at the spreading of waves, or to put it differently, the propagation of information during the RG-time evolution. Propagating modes correspond to

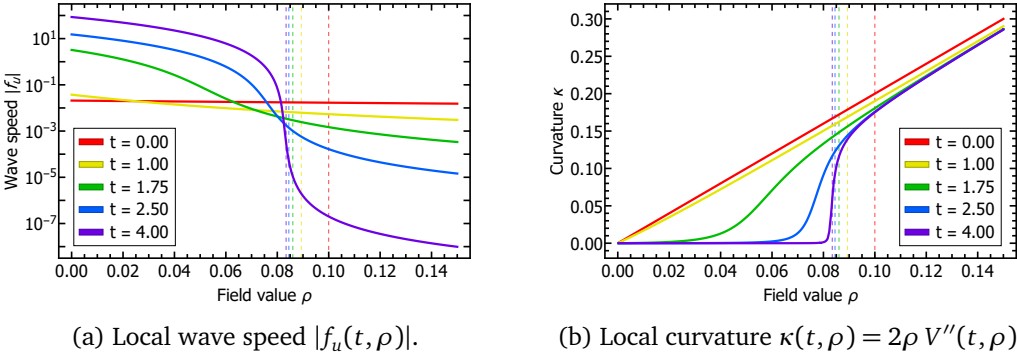

(a) Local wave speed $|f_u(t, \rho)|$.  (b) Local curvature $\kappa(t, \rho) = 2\rho\, V''(t, \rho)$.

Figure 6: Properties relating to the approach towards convexity for the initial values (33) with $\lambda_2 = -0.1$. The vertical dashed lines represent the position of the global minimum of the potential at the corresponding RG-time. The numerical simulation was performed with $K = 85$ elements and a local accuracy of order $N = 5$.

eigenvalues of the Jacobian of the system of conservation laws, which reduces in our case to $|\partial_u f(u)|$. The direction is always given to smaller field values, which naturally corresponds to the evolution direction from an RG perspective. Therefore, this quantity tells us at least qualitatively something about the locality of the RG-evolution in field space. From a technical perspective, the wave speed is an important quantity in our choice of the numerical flux, as it is directly related to the propagation of discontinuities. Additionally, it is relevant for the maximally allowed time step in explicit schemes to guarantee stability, see e.g. [48].

Turning back to our example case at the beginning of the section, i.e. (33) with $\lambda_2 = -0.1$ and $\lambda_4 = 1$, where we have used a local approximation of order $N = 5$ with 60 elements in the interval $0 \leq \rho \leq 0.15$ and 25 elements in the interval $0.15 \leq \rho \leq 1$. The locally resolved wave speed for different RG-times is shown in Figure 6a on a logarithmic scale. To guide the eye, the current minimum at each RG-time is indicated by a vertical dashed line. It is apparent that with progressing RG-time the wave speed splits into two domains, depending on the field value. For field values larger than the minimum, the wave speed is decreasing rapidly, i.e. it is decreasing exponentially fast. On the other hand, for field values smaller than the minimum, i.e. in the flat region of the potential, the wave speed is growing exponentially. A direct consequence is that explicit time steppers work extremely well in the non-flat region, because the time steps can be chosen increasingly larger as RG-time progresses, while in the flat region the time steps would be exponentially smaller and implicit methods are preferred. This comes

Table 1: Exact and reconstructed parameters of the power law behaviour (36) in the vicinity of the second order phase transition shown in Figure 4. The brackets indicate the $1\sigma$ uncertainty of the $\chi^2$-fit and the exact result is also given with numerical values for better comparability.

|  | Parameter | | |
|---|---|---|---|
|  | Prefactor | Critical exponent | Critical coupling |
|  | $\alpha$ | $\nu$ | $-\lambda_2^{\text{crit}}$ |
| Exact result | $\sqrt{2}$ | $\frac{1}{2}$ | $(6\pi^2)^{-1}$ |
|  | 1.4142 | 0.50000 | 0.01688686 |
| $\chi^2$-fit | 1.4161(16) | 0.50023(25) | 0.01688684(15) |

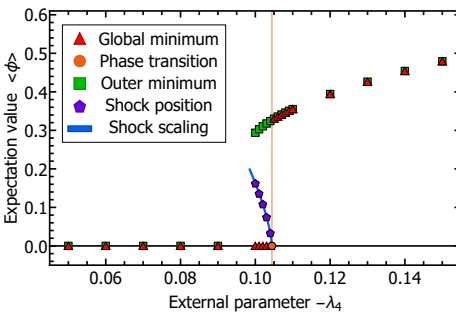
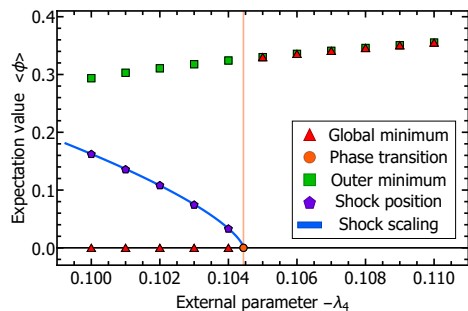

(a) Wider parameter range.  (b) Vicinity around the phase transition.

Figure 7: First order phase transition for the initial conditions (40). An extensive description can be found in the main text. The numerical simulation was performed with $K = 200$ elements and a local accuracy of order $N = 5$.

with implications for Taylor series methods, which are a popular choice in the fRG community, see e.g. [21, 23, 25], i.e. it provides an a posteriori justification for its use away from the flat region, due to the exponentially increasing locality of the solution. However, this should not be used as an a priori justification of its use. Similarly, Finite Difference based methods, see e.g. [9, 13, 14, 16–19, 26], will benefit from taking these considerations, especially the direction of the wave propagation, into account. Additionally, we would like to note that this analysis does not replace a proper stability analysis for these approaches, but simply provides an intuitive understanding with non-binding consequences.

As outlined previously, the separation of the solution at infinite RG-time into two regimes is closely linked to the flatness of the potential, i.e. its convexity. This also implies the vanishing of higher order couplings in the flat region. Therefore, the curvature

$$\kappa(\rho) = 2\rho\, V''(\rho) = 2\rho\, u'(\rho)\,, \tag{37}$$

provides a good measure for the flatness of the potential. The full curvature mass of the radial mode in $O(N)$-models is given by

$$m_{\text{curv}}^2 = u(\rho) + \kappa(\rho)\,, \tag{38}$$

more details can be found in Section 2. Therefore, a vanishing curvature (37) implies a vanishing curvature mass (38) of the radial mode in the flat region. The result for the curvature, in the same setting as the wave speed, is shown in Figure 6b. As for the wave speed, the minima at the shown RG-times are indicated by vertical dashed lines to guide the eye. The approach towards zero of the curvature in the flat region is clearly visible, similarly to the jump discontinuity that necessarily forms at the minimum. However, this discontinuity forms, just like the non-analytic point in the derivative of the potential itself, only in the asymptotic limit. Additionally, these findings are promising for future calculations in the $O(N)$-model at finite N, since the calculation of the curvature does not introduce new problems and is the only new ingredient entering at finite $N$. Within this setting we also do not expect a loss of accuracy despite the increasingly non-analytic behavior of the derivative. This is a clear advantage over pseudo-spectral methods, which are also designed to achieve high accuracy, put forward in [27–29]. They perform extremely well, if the solution is sufficiently smooth, however this is inherently not the case near phase transitions in the fRG. Additionally, it is worthwhile noting that these properties make pseudo-spectral approximation a good choice for the approximation of the momentum dependence of correlation function in Euclidean spacetime, see e.g. [74, 75]

### 3.2.3 Convergence

Due to the semi-analytic nature of the solution using the method of characteristics, c.f. Appendix B, we can benchmark the accuracy of our results obtained with the DG method. As with previous studies, we use the initial conditions (33) together with $\lambda_2 = -0.1$ and resolve the derivative of the effective potential over the interval $0 \leq \rho \leq 1$. The results are then compared at the RG-time $t = 4$. Hereby, we assume the result obtained via the method of characteristics to be the exact solution. Please note that this makes such a comparison for the situation with shocks considerably more complicated, which is why we refrain from considering it here. As explained in Section 2.2, we use an implicit solver for the time evolution. To avoid artificial enhancement of errors due to uncertainties thereof, we set the adaptive accuracy requirements close to machine precision. The results for the broken L2-norm between the two solutions for different orders of the local approximation order as a function of the number of elements, which are all equal in size, are shown in Figure 5. For our highest order of approximation $N = 5$ the results are only included for $K \leq 500$ elements, because the difference between the two results is at the level of the machine precision for more elements and a comparison is no longer insightful. The results are compatible with the expected power law like behavior for the convergence when increasing the number of elements $K$ and an exponential convergence when increasing the local approximation order $N$. To be more precise, we observe a behavior that can be parametrized as

$$\log_{10} ||u_h - u_{\text{exact}}||_{\Omega,h} = (a_1 + a_2 N) - (b_1 + b_2 N) \log_{10}(K).$$ (39)

In (39) we have temporarily restored the index $h$ again to denote the approximate solution. A $\chi^2$-fit to (39) is also shown in Figure 5 as solid lines, the parameters obtained are given in Table 2. In (39) the norm on the left-hand side denotes the broken L2-norm, i.e. the exact solution is projected to the same polynomial space as the numerical solution and the norm is then calculated elementwise therein and summed up. This result demonstrates the impressive convergence properties of the scheme.

## 3.3 First order phase transition

We now turn to the investigation of first order phase transitions, which have been investigated within the fRG in e.g. [9,24,58,76–80]. Including a $(\phi_a \phi^a)^3$ coupling into the classical action enables us to access a first order phase transition, see e.g. [56], which translates to the initial conditions

$$V_\Lambda(\rho) = \lambda_2 \rho + \frac{\lambda_4}{2}\rho^2 + \frac{\lambda_6}{3}\rho^3.$$ (40)

Similarly, to the second order case, c.f. Section 3.2, we fix all but one parameter and investigate the phase structure with respect to that parameter. To achieve a first order phase transition $\lambda_2$ and $\lambda_6$ need to be positive, while $\lambda_4$ needs to be negative. The initial values are chosen to

Table 2: Parameters obtained from a $\chi^2$-fit to (39) of the convergence behavior shown in Figure 5.

|  | | Parameter | | |
|---|---|---|---|---|
|  | $a_1$ | $a_2$ | $b_1$ | $b_2$ |
| $\chi^2$-fit | -6.0(10) | 2.22(31) | 1.36(40) | 1.34(12) |

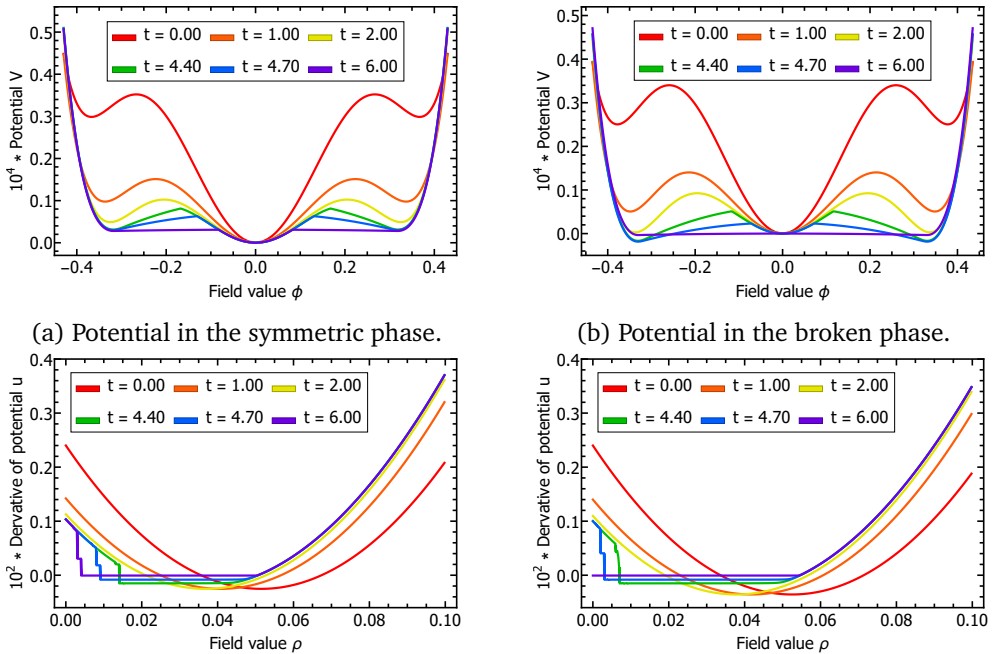

(a) Potential in the symmetric phase.

(b) Potential in the broken phase.

(c) Derivative of the potential in the symmetric phase.

(d) Derivative of the potential in the broken phase.

Figure 8: The effective potential $V(t, \rho)$ and its derivative $u(t, \rho)$ for two different values of the coupling $\lambda_4$ close to the first order phase transition shown in Figure 7. The numerical simulation was performed with $K = 200$ elements and a local accuracy of order $N = 5$. The results for the derivative of the potential were post-processed with a WENO limiter.

produce similar scales in the result compared to the results obtained in Section 3.2. Therefore, we keep $\lambda_4$ variable and set $\lambda_2 = 0.0024$, $\lambda_6 = 1$ to fixed values. Throughout this section we use $K = 200$ elements with a local approximation order of $N = 5$, with 150 elements distributed equally in $0 \leq \rho \leq 0.15$ and 50 elements distributed equally in $0.15 \leq \rho \leq 1$. The solution is obtained up to the RG-time $t = 6$, which was sufficiently large for all numerical simulations, i.e. the asymptotic result at infinite RG-time was obtainable via extrapolation if necessary.

A crucial difference between the initial conditions (33) and (40) concerns the monotonicity of the derivative of the effective potential at the initial scale, i.e. $u(0, \rho)$. While for the second order phase transition $u(0, \rho)$ was monotonically increasing as a function of $\rho$, in the case considered now, i.e. (40), it is not. To be more precise, it possesses a minimum for certain values of $\lambda_4 < 0$ and therefore a jump discontinuity will form as RG-time progresses. The underlying mechanism can easily be understood from the perspective of the characteristic velocity $\partial_u f(t, u)$, more details can be found in Section 3.1 and Appendix C. However, it is not clear whether the discontinuity forms in the physical relevant regime $\rho \in [0, \infty)$. Additionally, the results from Section 3.1 let us suspect that the shock will freeze in towards asymptotic RG-times. It turns out that this indeed happens and is the relevant mechanism behind the phase transition.

### 3.3.1 Phase structure I

We investigate the phase structure for the initial conditions (40), with the specific setup discussed around the equation. The resulting phase structure is shown in Figure 7, where a wider

range for the external parameter $\lambda_4$ is shown in Figure 7a and the vicinity around the phase transition is shown in Figure 7b. All quantities in the visualization of the phase structure are extrapolated to $t \to \infty$, the minima according to (35) and the final position of a possible jump discontinuity, i.e. shocks, is described later in detail. The outer minimum is depicted with green squares and the disappearance/jump to zero of it reflects the disappearance of the minimum in the initial conditions. However, the global minimum, depicted with red triangles, of the effective potential is either at $\rho = 0$ or agrees with the non-trivial, outer minimum at $\rho \geq 0$. A clear jump is visible where the potential switches between the symmetric and broken phase and the position is shown with a vertical orange line.

Before continuing the discussion of the phase structure and, in particular, the discussion of Figure 7b, it is instructive to look at the potential and its derivative at the two values of the coupling $\lambda_4$ which are closest to the phase transition, i.e. once in the broken phase and once in the symmetric phase, shown in Figure 8. Hereby we note that the results shown for the RG-time $t = 6$ are already sufficiently close to the infinite RG-time limit and for all discussions that follow we can treat them effectively as such. Focusing on the derivative of the effective potential $u(t, \rho)$, in both cases the appearance of a jump discontinuity is clearly visible. For a better depiction we have processed the result using a WENO limiter, following [81], removing the Gibbs oscillations around the shock. This is the origin of the flat looking pieces in the solution at the positions of shocks. However, the potential is obtained, as in Section 3.1, from the original data of the result. The two evolutions of the derivative of the potential, depicted in Figure 8c and Figure 8d, show a qualitative difference. In Figure 8c the position of the shock freezes and consequently the global minimum of the effective potential stays at $\rho = 0$ for all RG-times, see Figure 8a. This is in contradiction to the case depicted in Figure 8d, here the position of the shock moves to unphysical values and effectively flattens out the potential for all field values smaller than the outer minimum, making it the global minimum, depicted in Figure 8b.

### 3.3.2 Mechanism for a first order phase transition

The analysis above uncovers a potential mechanism for first order phase transitions: In the vicinity of the phase transition a cusp forms in the effective potential, or equivalently a shock in the derivative of the potential, during the RG-time evolution between two minima. The shock now propagates towards smaller field values and if the inner minimum was the preferred one before, the phase transition happens if the shock hits the inner minimum. The final position of the shock $\xi(t \to \infty)$ as a function of some external parameter, e.g. a coupling in the classical potential, temperature or chemical potential, can now be used to describe the phase transition equivalently. The propagation speed of the shock is dominantly driven by the values of the derivative of the potential at larger field values, c.f. (27). However, at roughly the same RG-time when the shock forms, the potential also starts to flatten, starting from the outer minimum, i.e. the potential approaches convexity. This process is triggered locally from the existence of a zero crossing in the derivative of the potential, and therefore independent of the global structure of the potential. Consequently, the propagation of the shock is dominantly driven by auxiliary, massless modes of the flat region and becomes at least partially insensitive to the details of the theory. This mechanism suggests a power law like behavior of the final position of the shock in the vicinity of the phase transition, which we will confirm for our current setting.

In our present case of the theory in the large $N$ limit, the formation of shock is guaranteed due to conservative form equation (25), combined with the non-monotonicity of the initial state. Therefore, the inner minimum is at $\rho = 0$ and the condition for the phase transition turn into $\xi(t \to \infty) = 0$.

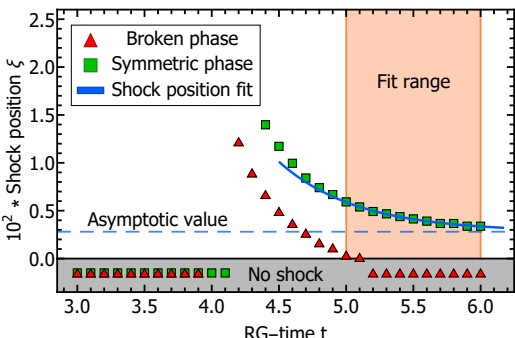

Figure 9: Positions of the shocks for the examples shown in Figure 8 together with the fit of the asymptotic behavior for the case shown Figure 8c.

Obviously, one should be cautious whether this mechanism generalizes to first order phase transitions in generic theories. We will comment on this at the end of this section, after finishing the discussion of the phase structure in our current setting. However, before continuing we would like to note that the propagation of a discontinuity in the vicinity of a first order phase transition has also been seen in [58], where the method of characteristics was used to resolve the phase structure of an NJL type model.

### 3.3.3 Phase structure II

Having identified the relevant mechanism for the phase transition shown in Figure 7, we can turn back to its description, including the final position of the shock, which are displayed with purple pentagons. It is now obvious that we get a good description of the phase structure in terms of the final position of the shock. To obtain the position of the discontinuity at infinite RG-time we follow the logic presented in Section 3.2, i.e. at large RG-times we expect an exponential decay

$$\xi(t) = \xi_{\text{final}} + a_\xi e^{-b_\xi t}, \quad \text{for} \quad t \gg 0. \tag{41}$$

This expectation is also supported by the asymptotic behavior extracted analytically from the Riemann problem, c.f. Section 3.1.1. To apply (41) we have extracted the position of the shock at 11 equally spaced points between the RG-times $t = 5$ and $t = 6$ using an appropriate concentration kernel, c.f. Appendix C, and then extracted the relevant information using a $\chi^2$-fit. The trajectories of the shocks from the evolutions shown in Figure 8 are depicted in Figure 9, which justifies the use of (41). Additionally, it should be noted that the trajectory with a finite final position of the shock shown in Figure 9 is the most extreme cases present, i.e. the exponential decay started at earlier RG-times for other values of the coupling with $\xi_{\text{final}} > 0$.

Following the discussion presented in Section 3.3.2, we expect a power law like behavior for the final position of the shock as a function of the coupling

$$\xi_{\text{final}} = \begin{cases} \beta \left| \lambda_4 - \lambda_4^{\text{crit}} \right|^\zeta, & \lambda_4 \geq \lambda_4^{\text{crit}}, \\ 0, & \lambda_4 < \lambda_4^{\text{crit}}. \end{cases} \tag{42}$$

Indeed, we find a very good agreement between the final positions of the shock and (42), the coefficients obtained from a $\chi^2$-fit are collected in Table 3. As for the second order phase transition, we obtain a very accurate estimate for the critical coupling, also shown with an orange circle in Figure 7. The critical exponent comes out at $\zeta = 0.683 \pm 0.013$ and it will be very interesting to investigate whether this value can be obtained from an associated fixed

point potential, which necessarily is either a partial fixed point or discontinuous, for a full study of the fixed points within this theory looking for continuous solutions see [82,83]. Non-analytic fixed point potentials have been found very recently [84] and it will be very interesting to explore the relation of our results to the ones presented therein, since the results share some qualitative features.

### 3.3.4  Generalization of the mechanism to other theories

It seems rather plausible that the mechanism outlined in Section 3.3.2 persists in general, at least to some extent. The first obvious generalization is to go beyond large $N$ and look at the flow equation (21) for finite $N$. Staying close to the conservative formulation employed so far, c.f. (25), we can express the flux for finite $N$ by inclusion of a diffusion term

$$f(t,u,\kappa) = f_{\text{Conv}}(t,u) + f_{\text{Diff}}(t,u,\kappa), \tag{43}$$

where the diffusion term depends additionally on the curvature defined in (37). DG schemes for diffusion terms are a well studied subject, see e.g. [85–87]. The first term on the right-hand side in (43) is the flux used in the large $N$ limit (24) and the additional term contains the contribution of the radial mode

$$f_{\text{Diff}}(t,u,\kappa) = -\frac{A_d}{N-1} \frac{(\Lambda e^{-t})^{d+2}}{(\Lambda e^{-t})^2 + u + \kappa}. \tag{44}$$

From a practical perspective (44) is a diffusion term, hence it has the possibility to smear out potential shocks. Away from any potential shocks this equation is still convection dominated, since the curvature appearing in the denominator is comparatively small. However, at field values around the shock it might give a sizeable contribution. However, in close proximity of the phase transition, i.e. when the shock, or a slightly smeared shock, approaches zero, it becomes important that (44) only depends on the curvature, which vanishes exactly at vanishing field value. Due to this reason, we expect a shock to be present in the direct vicinity of the phase transition. This marks a special regime at a first order phase transition, like the scaling regime at a second order phase transition. How this plays out in detail, especially in combination with the approach to convexity, will be extremely interesting to pursue in the near future. Particularly, the Péclet number, i.e. the convection over diffusion rate, might be a good start to quantify the competition between the different terms in (44).

Similarly, the presence of Fermions amounts to an additional source term in (43) in LPA. This potentially spoils the outlined mechanism in a trivial manner within this truncation. In this situation the phase transition is not fluctuation induced, but simply present due to the mean-field fermionic determinant, and an investigation should involve at least a field dependent Yukawa coupling to have the field dependent masses of fermions and bosons on the same footing. The field dependence of the Yukawa coupling in such theories was partially investigated in [88–92].

Table 3: Parameters obtained from a $\chi^2$-fit to (42) of the positions of the shocks depicted in Figure 7.

|  | Parameter | | |
|---|---|---|---|
|  | Prefactor $\beta$ | Critical exponent $\zeta$ | Critical coupling $-\lambda_4^{\text{crit}}$ |
| $\chi^2$-fit | 6.57(45) | 0.683(13) | 0.104438(28) |

Additionally, whether this mechanism can be used to extract properties of a first order phase transitions such as the nucleation rate in a convenient manner will be interesting to pursue.

# 4 Conclusion and Outlook

In this work, we have presented the applicability and advantages of applying Discontinuous Galerkin methods to the flow equations arising within the Functional Renormalization Group. As application, we considered the $O(N)$-model in the large $N$ limit in the Local Potential Approximation, where the flow equation of the effective potential can be cast into a conservative form, Section 2.2, which allows for a straightforward application of DG schemes. We considered the associated Riemann problem, as well as initial conditions that lead to a first or second order phase transition. The Riemann problem is considered in Section 3.1. It mainly led to the conclusion that shocks propagate only a finite range in field space. Therefore, they are still present in the solution at asymptotically large RG-times.

The case of a second order phase transition is presented in Section 3.2. We reproduced well known results from the literature and demonstrated in addition the expected convergence behavior of the scheme. The underlying stability and convergence properties also hold in the flat region of the potential, which contrasts with methods that rely on the smoothness of the solution, c.f. Section 3.2.2.

Initial conditions that lead to a first order phase transition are studied in Section 3.3. We discovered the formation of a shock in the derivative of the potential, leading to the mechanism behind first order phase transitions, explained in Section 3.3.2. This leads to an additional description of the phase structure in terms of the shock. In the vicinity of the phase transition, the position of the shock shows a power law behavior, like the order parameter in a second order phase transition.

These very promising results are the starting point for exciting follow-up projects. One part consists of investigating the mechanism for first order phase transitions further and establishing it in general. This also includes making a connection to the usual observables considered at such a phase transition. On the other hand, applying DG schemes to the PDE part of fRG equations is a promising route for reliable, precision calculations. Our results represent a very important step towards understanding the phase structure of strongly correlated systems such as QCD or the Hubbard model.

## Acknowledgments

We would like to thank Peter Bastian, Luca Del Zanna, Bernd-Jochen Schaefer for discussions and in particular Stefan Floerchinger and Jan M. Pawlowski for discussions and work on related topics.

**Funding information** This work is supported by the BMBF grant 05P18VHFCA, and is part of and supported by the DFG Collaborative Research Centre SFB 1225 (ISOQUANT).

## A Local approximation

In Section 1.2 we have introduced a dual expansion basis: $\psi_n(x)$ for the modes and $l_i^k(x)$ for the nodes, c.f. (5). The simplest, practical choice for the mode basis $\psi_n(x)$ is the set of

orthogonal Legendre polynomials $P_n$, which are part of the large family of Jacobi polynomials $P^{(\alpha,\beta)}$.

The Jacobi polynomials $P_n^{(\alpha,\beta)}(x)$ are the solution to the singular Sturm-Liouville problem

$$\frac{\mathrm{d}}{\mathrm{d}x}\left[(1-x^2)\omega(x)\frac{\mathrm{d}}{\mathrm{d}x}P_n^{(\alpha,\beta)}(x)\right]=-\lambda_n\omega(x)P_n^{(\alpha,\beta)}(x), \tag{A.1}$$

defined on the interval $[-1,1]$. In (A.1) $\omega(x)=(1-x)^\alpha(1+x)^\beta$ is the weight function and $\lambda_n=n(n+\alpha+\beta+1)$ are the eigenvalues. The Jacobi polynomials satisfy the weighted orthonormality relation

$$\int_{-1}^{1}\mathrm{d}x\;\omega(x)P_n^{(\alpha,\beta)}(x)P_m^{(\alpha,\beta)}=\delta_{nm}. \tag{A.2}$$

To construct the polynomials, it is convenient to use their recurrence relation, see e.g. [48], which relates the higher order $P_n$ to the lower ones,

$$xP_n^{(\alpha,\beta)}(x)=a_nP_{n-1}^{(\alpha,\beta)}(x)+b_nP_n^{(\alpha,\beta)}(x)+a_{n+1}P_{n-1}^{(\alpha,\beta)}(x), \tag{A.3}$$

where the coefficients are defined as

$$a_n=\frac{2}{2n+\alpha+\beta}\sqrt{\frac{n(n+\alpha+\beta)(n+\alpha)(n+\beta)}{(2n+\alpha+\beta-1)(2n+\alpha+\beta+1)}},$$
$$b_n=-\frac{\alpha^2-\beta^2}{(2n+\alpha+\beta)(2n+\alpha+\beta+2)}. \tag{A.4}$$

The recurrence relation can be used once the initial polynomials are defined,

$$P_0^{(\alpha,\beta)}(x)=\sqrt{2^{-\alpha-\beta-1}\frac{\Gamma(\alpha+\beta+2)}{\Gamma(\alpha+1)(\beta+1)}},$$
$$P_1^{(\alpha,\beta)}(x)=\frac{1}{2}P_0^{(\alpha,\beta)}(x)\sqrt{\frac{\alpha+\beta+3}{(\alpha+1)(\beta+1)}}\left[(\alpha+\beta+2)x+(\alpha-\beta)\right].$$

Derivatives can be computed from the lower order polynomials using the important relation

$$\frac{\mathrm{d}}{\mathrm{d}x}P_n^{(\alpha,\beta)}(x)=\sqrt{n(n+\alpha+\beta+1)}P_{n-1}^{(\alpha+1,\beta+1)}(x). \tag{A.5}$$

The Legendre polynomials are the special case $\alpha=\beta=0$, i.e. $P_n(x)=P_n^{(0,0)}$ and their properties and relations are easily obtained from the ones for the Jacobi polynomial.

In our implementation of the DG discretization method, we used the following convention for $\psi_n(x)$

$$\psi_n(x)=\sqrt{\frac{2n-1}{2}}P_{n-1}(x). \tag{A.6}$$

The nodal basis functions are chosen as the standard Lagrange interpolating polynomials,

$$l_i(x)=\prod_{j=1\;j\neq i}\frac{x-x_j}{x_j-x_i}, \tag{A.7}$$

which are well define and unique if the nodes $x_i$ are all distinct. It is advantageous to select the nodes such that the transformation matrix between the modal representation $\hat{u}_n$ and the nodal representation $u(x_i)$ is well conditioned. It can be shown, see e.g. [48], that the Legendre-Gauss-Lobatto (LGL) points, defined as the $N$ zeros of the equation

$$(1-x^2)P_N'(x)=0, \tag{A.8}$$

amount to an optimal choice.

# B Method of characteristics

In this appendix, we present the analytic solution of Equation (25). The PDE is a scalar quasi-linear partial differential equation in conservative form, therefore an implicit solution can be obtained using the method of characteristics [93]. The equation in conservative form is

$$\partial_t u(t,\rho) + \partial_\rho \left( A_d \frac{(\Lambda e^{-t})^{d+2}}{(\Lambda e^{-t})^2 + u(t,\rho)} \right) = 0 \,, \tag{B.1}$$

and can be express in a quasilinear form performing the derivative on the flux,

$$\partial_t u(t,\rho) - A_d \frac{(\Lambda e^{-t})^{d+2}}{\left[(\Lambda e^{-t})^2 + u(t,\rho)\right]^2} \partial_\rho u(t,\rho) = 0 \,, \tag{B.2}$$

combined with the initial condition

$$u(0,\rho) = u_0(\rho) \,. \tag{B.3}$$

The solution can be found by introducing the characteristic curves that are the solution of

$$
\begin{aligned}
\frac{dt(s)}{ds} &= 1 \,, \\
\frac{d\rho(s)}{ds} &= -A_d \frac{(\Lambda e^{-t(s)})^{d+2}}{\left[(\Lambda e^{-t(s)})^2 + u(s)\right]^2} \,, \\
\frac{du(s)}{ds} &= 0 \,,
\end{aligned}
\tag{B.4}
$$

combined with the initial conditions

$$
\begin{aligned}
t(0) &= 0 \,, \\
\rho(0) &= \rho_0 \,, \\
u(0) &= u_0(\rho_0) \,.
\end{aligned}
\tag{B.5}
$$

This system of ordinary differential equation is equivalent to the original partial differential equation (B.2) if we define

$$u(s) = u(t(s), \rho(s)) \,. \tag{B.6}$$

The system (B.4) can easily be integrated, noting that $u(s)$ is constant along the characteristic and $t(s)$ is the curve parameter. The result can be written as

$$u(s) = u_0(\rho_0) \,, \quad \text{and} \quad t(s) = s = t \,, \tag{B.7}$$

and

$$\rho(t) = \rho_0 - A_d \int_0^t \frac{(\Lambda e^{-s})^{d+2}}{\left[(\Lambda e^{-s})^2 + u_0(\rho_0)\right]^2} ds \,. \tag{B.8}$$

The integral can be carried out, leading to

$$
\begin{aligned}
\rho(t) = \rho_0 - \frac{A_d \Lambda^d}{2} \Bigg[ &\frac{e^{-dt}}{u_0(\rho_0) + (\Lambda e^{-t})^2} - \frac{1}{u_0(\rho_0) + \Lambda^2} \\
&- \frac{d(\Lambda e^{-t})^{d-2}}{(d-2)\Lambda^d} {}_2F_1\left(1, \frac{2-d}{2}, \frac{4-d}{2}, -\frac{u_0(\rho_0)}{(\Lambda e^{-t})^2}\right) \\
&+ \frac{d\Lambda^{-2}}{d-2} {}_2F_1\left(1, \frac{2-d}{2}, \frac{4-d}{2}, -\frac{u_0(\rho_0)}{\Lambda^2}\right) \Bigg] \,,
\end{aligned}
\tag{B.9}
$$

where $_2F_1$ is the Gaussian or ordinary hypergeometric function, see e.g. [94]. The equation (B.9) is a transcendental equation between $\rho_0$, the position at the initial RG-time where $u$ has the value $u_0(\rho_0)$ and $\rho$, which is the position at RG-time $t$ where $u$ has the same value. Formally this can now be inverted, obtaining

$$\rho_0 = \rho_0(t, \rho), \tag{B.10}$$

giving the initial position of the value $u_\Lambda(\rho_\Lambda)$ as a function of the final one $\rho(t)$. The solution can be constructed using this inverse function as

$$u(t, \rho) = u_0(\rho_0(t, \rho)). \tag{B.11}$$

Practically, except for very simple cases, the solution of the transcendental equation (B.9) cannot be achieved analytically. Therefore, the inversion is performed numerically.

The equation (B.9) can be used to find a simple expression for the RG-time evolution of the minima of the potential, indeed if one use that $u_0(\rho_0) = 0$ and hence $_2F_1 = 1$, we obtain

$$\rho_{\min}(t) = \rho_{\min}(0) - \frac{A_d \Lambda^{d-2}}{d-2}\left[1 - e^{-(d-2)t}\right]. \tag{B.12}$$

## C Shock propagation and detection

### C.1 Position of the shock

Consider an interval $[\rho_L, \rho_R]$ that contains the position of the discontinuity at a given RG-time $t$, namely $\rho_L \le \xi(t) \le \rho_R$. Additionally, the interval must be chosen small enough that it only contains a single discontinuity. If this is not the case, it can always be split into multiple intervals. The integral in $\rho$ − space of our equation of interest (25) on this interval is

$$\frac{\mathrm{d}}{\mathrm{d}t}\int_{\rho_L}^{\rho_R}\mathrm{d}\rho\, u(\rho, t) - \int_{\rho_L}^{\rho_R}\mathrm{d}\rho\, \partial_\rho f(t, u(\rho)) = 0. \tag{C.1}$$

Splitting the integral around the discontinuity results in

$$\frac{\mathrm{d}}{\mathrm{d}t}\int_{\rho_L}^{\xi(t)}\mathrm{d}\rho\, u(\rho, t) + \frac{\mathrm{d}}{\mathrm{d}t}\int_{\xi(t)}^{\rho_R}\mathrm{d}\rho\, u(\rho, t) = f(t, u(t, \rho_R)) - f(t, u(t, \rho_L)). \tag{C.2}$$

The RG-time derivative can be done explicitly and leads to

$$\begin{aligned}
\frac{\mathrm{d}\xi(t)}{\mathrm{d}t}(u(\rho_R, t) - u(\rho_L, t)) &- f(t, u(t, \rho_R)) + f(t, u(t, \rho_L)) \\
&= -\int_{\xi(t)}^{\rho_R}\mathrm{d}\rho\, \partial_t u(\rho, t) - \int_{\rho_L}^{\xi(t)}\mathrm{d}\rho\, \partial_t u(\rho, t).
\end{aligned} \tag{C.3}$$

In the limit $\rho_L \to \xi^-(t)$ and $\rho_R \to \xi^+(t)$ the right-hand side vanishes, and we obtain the equation

$$\frac{\mathrm{d}\xi(t)}{\mathrm{d}t}(u_R(t) - u_L(t)) - f_R(t) + f_L(t) = 0, \tag{C.4}$$

where we have used the definitions

$$
\begin{aligned}
u_{\mathrm{R}}(t) &= \lim_{\rho \to \xi^+(t)} u(\rho, t), \\
f_{\mathrm{R}}(t) &= \lim_{\rho \to \xi^+(t)} f(t, u(\rho, t)), \\
u_{\mathrm{L}}(t) &= \lim_{\rho \to \xi^-(t)} u(\rho, t), \\
f_{\mathrm{L}}(t) &= \lim_{\rho \to \xi^-(t)} f(t, u(\rho, t)).
\end{aligned}
\tag{C.5}
$$

The equation for the position of the discontinuity is described by

$$
\frac{\mathrm{d}\xi(t)}{\mathrm{d}t} = \frac{f_{\mathrm{R}}(t) - f_{\mathrm{L}}(t)}{u_{\mathrm{R}}(t) - u_{\mathrm{L}}(t)} = \frac{[\![f]\!]}{[\![u]\!]},
\tag{C.6}
$$

which can be integrated to obtain the RG-time evolution of the shock.

## C.2 Shock detection

To determine the position of the jump discontinuities in our numerical approximation $u_h(t, \rho)$ we follow the procedure outlined in [95, 96], i.e. the method of concentration.

We briefly summarize here how this procedure is practically applied. While shock capturing schemes are very interesting by itself and are a promising future direction, we restrict ourselves here to the extraction of the position of discontinuities during post-processing. Discontinuities, i.e. their position and height can be extracted by folding the function $f(x)$, which is assumed to be piecewise continuous, with a suitable concentration kernel, which acts as

$$
K_\varepsilon * f(x) = [\![f]\!](x) + \mathcal{O}(\varepsilon).
\tag{C.7}
$$

To define the concentration kernel from a numerical point of view, we have to understand how a discontinuous function is expanded in our basis. Consider the expansion of a piecewise smooth function $f(x)$ in terms of Jacobi polynomials,

$$
f(x) \simeq \sum_{k=0}^{N} \hat{f}_k P_k(x), \quad \text{with} \quad \hat{f}_k = \int_{-1}^{1} \mathrm{d}x\, \omega(x) f(x) P_k(x).
\tag{C.8}
$$

Utilizing the Sturm-Liouville equation (A.1), assuming that the function $f$ has a jump $[\![f]\!](c)$ for $x = c$, it is possible to obtain an estimation for the decay of the spectrum $\hat{f}_k$ with $k$,

$$
\begin{aligned}
\hat{f}_k &= \frac{-1}{\lambda_k} \int_{-1}^{1} \mathrm{d}x \left[(1 - x^2)\omega(x)P_k'(x)\right]' f(x) \\
&= [\![f]\!](c)\frac{1}{\lambda_k}(1 - c^2)\omega(c)P_k'(x) + \mathcal{O}\left(\frac{1}{\lambda_k}\right).
\end{aligned}
\tag{C.9}
$$

This equation expresses the fact that next to a jump the coefficients of the mode expansion decays like $\frac{1}{\lambda_k}$, which is substantially slower than far away from a jump. In (C.9) $\lambda_k$ refers to the eigenvalue of the associated Sturm-Liouville equation, c.f. (A.1). Motivated by this characteristic property of the spectrum of a particular polynomial expansion, it is possible to define a quantity that detects the discontinuity from the mode expansion of the function. The concentration kernel for Legendre polynomial was obtained in [95] and is defined as

$$
K_N^\sigma * f = \sqrt{2}\frac{\pi}{N}\sqrt{1 - x^2} \sum_{k=1}^{N} \sigma\left(\frac{|k|}{N}\right) \hat{f}_k \psi_k'(x),
\tag{C.10}
$$

where $\sigma(\xi)$ is an adequate concentration factor. There are different possibility for this function and an extensive discussion can be found in [95]; for our implementation we have made the simple choice of $\sigma(\xi) = 1$. In the vicinity of the discontinuity, and away from it, this kernel behaves as

$$K_N^\sigma * f = \begin{cases} \mathcal{O}(\frac{1}{N}), & x \neq c \,, \\ [\![ f ]\!](c) + \text{const.} \frac{\log N}{N} \,, & x = c \,. \end{cases} \tag{C.11}$$

Consequently, it is possible to pinpoint the discontinuity, when examining the scaling of this operator with the number of nodes. However, is more convenient to enhance the separation of scale between the smooth part and the discontinuity, namely

$$N^{\frac{p}{2}}(K_N^\sigma * f)^p = \begin{cases} \mathcal{O}\left( N^{-\frac{p}{2}} \right), & x \neq c \,, \\ [\![ f ]\!](c) N^{\frac{p}{2}} \,, & x = c \,, \end{cases} \tag{C.12}$$

where $p$ is the enhancement exponent. Using this operator, it is possible to construct an operator that is non-vanishing only in the presence of the jump,

$$K_{N,J}^p * f = \begin{cases} K_N * f \,, & \text{if } \left| N^{\frac{p}{2}}(K_N^\sigma * f)^p \right| > J \,, \\ 0 \,, & \text{otherwise,} \end{cases} \tag{C.13}$$

where $J$ is an appropriately chosen threshold. This additional definition becomes very important for smaller values of $N$ if we want to achieve a good separation of scales between shocks and smooth parts of the solution. In our implementation, we have chosen the heuristic values $p = 2$ and $J = 5.0 \times 10^{-8}$. With this set of parameters, we were able to detect the discontinuities in Section 3.3 efficiently.

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
