# Peer review of "Resolving phase transitions with Discontinuous Galerkin methods"

_SciPost Physics, doi:SciPost Phys. Core 6, 071 (2023)_

## Round 2 · Referee Report · Anonymous · 2019-6-17

Strengths

1. The paper seeks to introduce and discuss the applicability of Discontinuous Galerkin
methods in solving functional renormalization group flow equations. This could in principle be a useful goal.

Weaknesses

1. Although the basic goal of introducing Discontinuous Galerkin methods to a new audience
could be useful if achieved, it is not clear that the results would be significant enough for
this journal.
2. Unfortunately the introduction to Discontinuous Galerkin methods is too brief to be
understood without already having knowledge of these methods, although it is possible to
fish out some introductory material buried much later in the paper.
2. The paper is applied only to the large-N O(N) models, apart from some brief speculations
at the end of sec. III. Such models are very special - in fact exactly soluble in this context also as the authors themselves review in appendix B. One is left with the impression that these
methods are only applicable to such first order non-linear equations, as appears also to be
true of methods based on a weak solution on which these methods appear to be closely
based. In this case, one is already better off with using the method of appendix B coupled with numerical inversion.
3. The number of typos, mis-spellings, grammatical inaccuracies and other evidences of lack
of care, is so great that it would be impractical to list more than a sample.
4. There are a number of places where the mathematical development and/or physical
reasoning seems also to be suspect.

Report

The authors' stated intention is "to close the cap [sic: should be "gap"] between the FRG and DG community". As someone from the FRG community, I found sec. IB far too brief to achieve this ambition. Even before this introduction starts there is reference to "conservation
equations" which one gradually realises means something quite specific here (normal use of
the word "conservation" would imply something -like a charge- is held constant) and to a
"weak formulation". These terms are never explained (to be joined by many others e.g. "CFL" conditions and "BDF" methods on p5, NB next sentence: "persevering"->preserving). On the other hand above eqn (23) you actually write "the first derivative of the potential is a
conserved quantity". At face value you seem to be telling me that the object in eqn (23) is a
constant with respect to t? Reference is made to intuition about relative rates of diffusion
versus convection, but the reader is never inducted into this way of thinking i.e. what precisely
this means for the FRG equation and how it relates to physics the reader from the FRG
community may have seen before.

The derivation of eqn (9) is given as integrating by parts eqn (7) to get (8) and then
integrating by parts back again. Since (9) is different from (7) there is clearly more to this, but readers are left to try to figure out why for themselves, or go learn about these methods from
somewhere where it is better explained. Eqn (10) is pulled out of a hat (NB "state the we work" -> state that we work), together with (11) (now in general dimension and not one dimension as earlier? why? - given that the rest of the paper is in one dimension).

Another example: eqn (27) on p5 is not explained, just quoted. Then it turns out it is derived
at the end of appendix C1! This can easily be missed unless one picks up on the one reference
on another matter at the end of sec. IIC on p11. Equation (27) is integrated in eqn (28). It
would appear that the authors assume that $u_L$ and $u_R$ are held constant, although the authors are silent on this point. When is such an assumption justified? (Incidentally in the next
paragraph I am not sure what word you actually meant by "prepositions").

Some more examples: eqn (35) is claimed to be generic for any coupling. Maybe so in large N but not generally. (Next section "It is instructing" -> It is instructive.) Why are Legendre
polynomials used? (App A. NB "Strum-Liouville"-> Sturm-Liouville, here & throughout.)

In many places two different forms of a sentence appear together, such as "are given can
easily be obtained", "convenient [with] respect compared to others" in App A. Many verbs
incorrectly conjugated e.g. "chose"-> choose, "condense" -> condensed, "outline" ->
outlined etc. Wrong words e.g. "underling" -> underlying, "limes" -> limits etc.

The authors need to go through the paper line by line e.g. using an editor that will prompt for all the grammatical inaccuracies. With some substantial improvements, not only to this but
the whole logic and presentation, this paper might find a useful home in a more pedagogically
minded journal. Sifting through this paper in its present form, I saw no evidence to make me
think that the work is suitable for this journal.

Requested changes

Not applicable

  • validity: low
  • significance: low
  • originality: ok
  • clarity: poor
  • formatting: good
  • grammar: mediocre

Author:  Nicolas Wink  on 2019-07-03  [id 553]

(in reply to Report 1 on 2019-06-17)
Category:
answer to question
objection

General points

Before addressing the individual criticism in the report we would like to outline two points:

  1. Despite being depicted as such in the report, the paper is neither meant to be overly pedagogical nor an introduction to the topic. First and foremost, it is a research paper with a slightly longer introduction chapter than usual.
  2. The physics aspect of the paper is not mentioned a single time in the report.

Weaknesses

Although the basic goal of introducing Discontinuous Galerkin methods to a new audience could be useful if achieved, it is not clear that the results would be significant enough for this journal.

This is not intended goal of paper. However, we are aware of the fact that these methods are not well known in the FRG community. Therefore, we included an introduction to the topic. We do think that the research content of the paper is well suited for this journal.

Unfortunately the introduction to Discontinuous Galerkin methods is too brief to be understood without already having knowledge of these methods, although it is possible to fish out some introductory material buried much later in the paper.

The introduction only aims at introducing DG schemes. However, we do agree that the section about numerical fluxes might be a bit short if the reader is not familiar with Finite Volume Methods. Therefore, we have extended this part. We do want to stress however, that we do not aim at giving a broad introduction to the topic. As a direct result, quite some of the technical details are in the Appendix.

The paper is applied only to the large-N O(N) models, apart from some brief speculations at the end of sec. III. Such models are very special - in fact exactly soluble in this context also as the authors themselves review in appendix B. One is left with the impression that these methods are only applicable to such first order non-linear equations, as appears also to be true of methods based on a weak solution on which these methods appear to be closely based. In this case, one is already better off with using the method of appendix B coupled with numerical inversion.

  • For the development of numerical methods it is extremely useful and customary to first investigate problems that are either solvable (semi)-analytically or by other means.
  • We would like to point out that the method of characteristics is not "better" than the scheme presented here. As a matter of fact, for quite some aspects presented in the paper DG schemes are computationally cheaper, e.g. the resolution of the RG-time evolution.
  • The order of the equation and the concept of a weak solution are entirely unrelated.
  • DG schemes are by no means restricted to first order differential equations. Since these schemes are almost never blindly applicable to new physical systems it is natural to start at the simplest possible approximation. A prominent use case for these schemes are the compressible Navier–Stokes equations, i.e. a second order non-linear set of equations. Additionally, we have added some references where diffusion equations have been studied with DG schemes.

The number of typos, mis-spellings, grammatical inaccuracies and other evidences of lack of care, is so great that it would be impractical to list more than a sample.

We agree and correct as many as possible.

There are a number of places where the mathematical development and/or physical reasoning seems also to be suspect.

We cannot comment to this general criticism, since the corresponding points are not apparent in the report and we obviously disagree.

Report

The authors' stated intention is "to close the gap...

This is our intention for the existence of the relevant subsections in the introduction, not the paper.

Even before this introduction starts there is reference to "conservation equations"...

It does not describe something quite specific in this context. The usual, general definition of a conservation law is $\nabla_\mu a^\mu = 0$. It is simply written slight different here. The term conservation law must not be confused with a locally conserved charge.

...These terms are never explained (to be joined by many others e.g. "CFL" conditions and "BDF"...

Not explained at length as we expect the reader to be familiar with the basics of numerical methods for partial differential equations. However, we have extended the existing qualitative explanation of the CFL condition slightly. Written out the acronym for the family of Backward differentiation formula (BDF) methods. We have erased the unnecessary reference to the weak solution before it was introduced.

...At face value you seem to be telling me that the object in eqn (23) is a constant with respect to t?

No, as usual for a conservation law (think of it as e.g. mass conservation for incompressible fluids), it is locally conserved and the rate of change is given by the flux. As pointed out above, while conservation laws and conserved charges are related, they must not be confused with each other.

Reference is made to intuition about relative rates of diffusion versus convection...

We do give some explanations in the introductions (RG flow as convection) and in Section 2.3. We have included an additional cross-reference at the first occurrence and extended the explanation slightly to make the analogy more apparent.

The derivation of eqn (9) is given as integrating by parts eqn (7) to get (8) and then...

While we do not aim at a general introduction to the topic of numerical fluxes, we agree that the explanation given might be to short and have extended the part correspondingly.

now in general dimension and not one dimension as earlier?...

It is still in one dimension, the vector notation is simply a lot more convenient to write the expressions. We have added the explicit expressions for one dimension to avoid confusion.

Another example: eqn (27) on p5 is not explained, just quoted...

We have added the missing cross-reference to the Appendix.

It would appear that the authors assume that $u_L$ and $r_R$ are held constant...

The paper states above (26) that piecewise constant initial conditions define the problem. This assumption holds by construction of the problem. Nevertheless, we added a note that the solution stays trivial piecewise constant for a propagating shock.

eqn (35) is claimed to be generic for any coupling...

We added a disclaimer for Large N theories.

Why are Legendre polynomials used?

Simplest and probably by far the most popular choice for an orthogonal basis in one dimension.

In many places two different forms of a sentence appear together...

We apologize and correct as many errors as possible.

---

## Round 3 · Referee Report · Anonymous (Referee 2) · 2019-8-1

Strengths

1) The paper introduces discontinuous Galerkin methods into the context of the FRG. This will prove very useful since we are often dealing with equations which develop discontinuities in some derivative, and a good numerical treatment is key for both qualitatively and quantitatively correct results.

2) Results on the O(N) model in the large N limit can be benchmarked against the analytic solution, which illustrates the stability of the numerical methods.

3) The paper studies both first and second order phase transitions, which spans virtually all cases of phase transitions one encounters. Interesting results on shock formation and rarefaction waves are presented.

Weaknesses

1) It would have been interesting to push the numerics to larger values of the RG time to check how long the numerical error is under control, especially in the case where shocks form.

2) As the first referee also said, the language level is mediocre, there are still a lot of typos/grammatical inaccuracies etc.

Report

Overall, I disagree with referee 1 - I find the introduction of the DG methods into the FRG context performed in the paper worthwhile and successful. The O(N) model in the large N limit is a well-known and analytically solvable testing grounds where numerical techniques can be tested in a controlled way. Of particular interest is the study of the formation of shocks and rarefaction waves, which is, to my knowledge, new in the FRG context. The paper and the methods it introduces will thus certainly make an impact on future research when it comes to studying phase transitions that feature non-smooth behaviour.

Having said that, there are some points that I would like the authors to address before I can recommend that the paper can be published.

  • As a minor suggestion, it might be useful to use $\tau$ instead of $t$ for (minus) the RG time to avoid confusion.
  • In section IA, where the authors discuss different approximation schemes, it seems fair to me to refer to the BMW scheme which resolves both momentum and field dependence, see e.g. [Phys.Lett. B632 (2006) 571-578].
  • It might help the inexperienced reader to point out more explicitly what the Galerkin part (eq. 7) and the discontinuous part of the scheme is.
  • Before eq. 10, a reference for the "Local Lax-Friedrichs flux" might be useful.
  • In section II, for the examples it should be clarified which dimensions they correspond to (Standard Model: d=4, condensed matter: typically d=3) and some references might be added.
  • Section IIA, second sentence: the action is expanded in powers of gradients of the field, not the field in terms of gradients.
  • The tensor structure of the regulator should be specified.
  • Before eq. 20, I suspect that $(t,\rho)$ should read $V(t,\rho)$.
  • CFL should be spelled out once.
  • Fig. 5 is discussed after Fig. 6, they should thus be reordered.
  • Fig. 5 shows the convergence of the numerical scheme for different numbers of elements and orders. In the text it says that this is taken at $t=1.75$, at the onset of the flattening of the potential. In this regime it is clear that the accuracy is very high since the potential is still smooth enough, and standard pseudo-spectral methods should work equally well up to this point. It would be more honest to report on the numerical error at the end point of the flow at $t=4$. (*)
  • Fig. 8c) and d): there seem to be numerical artefacts near vanishing field - these should be explained somewhere. (*)
  • Right before section IV: the authors mention the impact of fermions e.g. by a field dependent Yukawa coupling. Some references here seem to be adequate, e.g. [Phys.Rev. D94 (2016) no.3, 034016, Phys. Rev. D 91, 125003 (2015), Phys. Rev. B 94, 245102 (2016), Eur.Phys.J. C77 (2017) no.11, 743].
  • The authors use Legendre polynomials. While these provide exponential convergence for smooth functions, the subleading part of the rate of convergence is worse than that of Chebyshev polynomials, see [Boyd: Chebyshev and Fourier Spectral Methods, Chapter 2.13]. Why do the authors nevertheless use Legendre polynomials?
  • Eqs. C1 and C2: it seems that coming from the first to the second equation, the authors have employed the fundamental theorem of calculus. This is however only valid for continuous integrands. In general, the flux might contain a discontinuity, inherited from its dependence on u. This would give rise to additional terms in eq. C2. Can the authors elaborate on why these terms generically don't appear? (*)
  • Eq. C12: it wasn't clear to me what $\lambda_k$ refers to in this equation.
  • Last but not least, the manuscript still contains an awful lot of spelling mistakes and grammatical errors. I can only recommend (as referee 1) to carefully read and improve the language of the paper (main text and appendices) to improve its readability.

Requested changes

See report. The most important points are marked with a (*).

  • validity: high
  • significance: high
  • originality: good
  • clarity: good
  • formatting: excellent
  • grammar: below threshold

Author:  Nicolas Wink  on 2022-08-16  [id 2729]

(in reply to Report 1 on 2019-08-01)
Category:
answer to question
correction

Weaknesses

It would have been interesting to push the numerics to larger values of the RG time to check how long the numerical error is under control, especially in the case where shocks form.

We have checked this in the given setting of the second order phase transition, the convergence properties are independent of the final RG-time. However, a relative comparison between different RG-times would mostly check the convergence properties of the external library we used for solving the resulting set of odes, as stated in the paper. We don't think it is worth repeating the study for a solution with a shock, because obtaining a reference solution that is suitable for comparison comes with considerable amount of work. After all, the actual solution is also obtained numerically when using the method of characteristics. We added a statement clarifying this situation in the text.

As the first referee also said, the language level is mediocre, there are still a lot of typos/grammatical inaccuracies etc.

We corrected as many typos and grammatical inaccuracies as possible.

Report

As a minor suggestion, it might be useful to use $\tau$ instead of $t$ for (minus) the RG time to avoid confusion.

We would prefer to keep the current notation.

In section IA, where the authors discuss different approximation schemes, it seems fair to me to refer to the BMW scheme which resolves both momentum and field dependence, see e.g. [Phys.Lett. B632 (2006) 571-578].

The suggested paper is already cited, categorized as vertex expansion. Most of the works that are considered as vertex expansion also resolve the field dependence to at least some extent. We have changed the wording slightly to reflect better how these terms are understood.

It might help the inexperienced reader to point out more explicitly what the Galerkin part (eq. 7) and the discontinuous part of the scheme is.

We added a sentence explaining this.

Before eq. 10, a reference for the "Local Lax-Friedrichs flux" might be useful.

We added the original reference.

In section II, for the examples it should be clarified which dimensions they correspond to (Standard Model: d=4, condensed matter: typically d=3) and some references might be added.

We extended this part appropriately.

We extended this part appropriately.

Section IIA, second sentence: the action is expanded in powers of gradients of the field, not the field in terms of gradients.

Fixed the typo.

The tensor structure of the regulator should be specified.

We clarified that we used a diagonal regulator.

Before eq. 20, I suspect that $(t,\rho)$ should read $V(t,\rho)$.

Fixed the typo.

CFL should be spelled out once.

We included the full name once.

Fig. 5 is discussed after Fig. 6, they should thus be reordered.

This was done for styling reasons on purpose.

Fig. 5 shows the convergence of the numerical scheme for different numbers of elements and orders. In the text it says that this is taken at t = 1.75, at the onset of the flattening of the potential. In this regime it is clear that the accuracy is very high since the potential is still smooth enough, and standard pseudo-spectral methods should work equally well up to this point. It would be more honest to report on the numerical error at the end point of the flow at t=4.

We fully agree that it makes no sense to have this comparison at t=1.75. We updated the figure and the coefficient table to the comparison at $t=4$. The coefficients still agree within error tolerances, except for $a_1$. This is expected at different RG-times, since it simply accounts for the accumulated error from the RG-time integration.

Fig. 8c) and d): there seem to be numerical artefacts near vanishing field - these should be explained somewhere.

This is a result of the WENO limiter we used in order to remove the Gibbs oscillations from the plot. We have added a sentence clarifying this.

Right before section IV: the authors mention the impact of fermions e.g. by a field dependent Yukawa coupling. Some references here seem to be adequate, e.g. [Phys.Rev. D94 (2016) no.3, 034016, Phys. Rev. D 91, 125003 (2015), Phys. Rev. B 94, 245102 (2016), Eur.Phys.J. C77 (2017) no.11, 743].

We added references, in particular also [Phys.Rev. D90 (2014) no.7, 076002], which seems to be the first work considering this.

The authors use Legendre polynomials. While these provide...

For technical convenience since it only effects the subleading part and does not change any fundamental properties of the scheme. Additionally, they are the standard choice in Finite Element based approaches.

Eqs. C1 and C2: it seems that coming from the first to the second equation, ...

The integrands in C2 are continuous by design and an additional term arises from the discontinuity that's present, which can be seen in C3. We added a sentence clarifying that the interval needs to chosen sufficiently small such that only one discontinuity is present.

Eq. C12: it wasn't clear to me what $\lambda_k$ refers to in this equation.

We clarified this, it's the eigenvalue of the Sturm-Liouville equation, introduced below A1.

Last but not least, the manuscript still contains an awful lot of spelling mistakes and grammatical errors. I can only recommend (as referee 1) to carefully read and improve the language of the paper (main text and appendices) to improve its readability.

We corrected as many typos and grammatical inaccuracies as possible.

Attachment:

dc_galerkin_frg.pdf

---

## Round 3 · List of Changes

• Extended explanation of numerical flux
  • Slight extension of several other explanations
  • Fixed typos

---

## Round 4 · Referee Report · Anonymous · 2023-7-2

Strengths

As outlined in previous reports, the strengths are that the paper

1. Introduces discontinuous Galerkin methods into the context of the FRG.

2. Results on the O(N) model in the large N limit can be benchmarked against the analytic solution, which illustrates the stability of the numerical methods.

3. Many interesting insights resulting from numerical and analytical investigations are formulated and will be of interest to a large community of researchers.

Weaknesses

1. The main weakness is for me the restriction to the large-N limit which is clearly an idealization for any real physics system. But given that the paper contains anyway enough new material this is acceptable.

Report

The paper has gone through several stages of refereeing and was significantly improved during the process. It is overall very interesting and a substantial step forward for the aim to find reliable numerical methods to solve functional renormalization group equations. In fact, it has already triggered a set of new developments in the field, despite the fact that it is until now only available as a preprint.

I would therefore like to recommend publication of the manuscript in SciPost Physics.

Requested changes

I have just one suggestions that the authors may take into account: The formulation of the flow equation for the effective potential in the form of a local conservation law plays a significant role for the whole paper. However, the physical significance of this conservation law is not discussed very much. I would find it useful to have also an integrated form (with respect to the field space variable rho) stated and briefly discussed.

Here are some typos that came to my attention:
page 3: the numerical fluxes are define ... -> ...are defined...
page 4: an immense variety of physical system at different energy scale... -> ... physical systems at different energy scales...
page 5: stability preserving scheme Runge-Kutta scheme -> stability preserving Runge-Kutta scheme

  • validity: high
  • significance: high
  • originality: high
  • clarity: high
  • formatting: good
  • grammar: reasonable

Author:  Nicolas Wink  on 2023-07-10  [id 3792]

(in reply to Report 1 on 2023-07-02)

Hi,

we thank the referee for the report.
While we do agree that some comments regarding the interpretation of conservation may have been appropriate in the original manuscript, we do not think that it is timely anymore.
The question has been addressed to a various degree in quite some follow-up publications:

Grossi, Ihssen, Pawlowski, Wink - Phys.Rev.D 104 (2021)
Koenigstein et al - Phys.Rev.D 106 (2022) 6, 065012
Koenigstein et al - Phys.Rev.D 106 (2022) 6, 065013
Steil, Koenigstein - Phys.Rev.D 106 (2022) 6, 065014
Ihssen, Pawlowski, Sattler, Wink - https://arxiv.org/abs/2207.12266
Ihssen, Sattler, Wink - Phys.Rev.D 107 (2023) 11, 114009

There are more, but these should be the most relevant ones for this question.
The integrated form itself is not very illuminating, since the equation is derived by taking a derivative and the integral simply yields the original flow equation [e.g. (22)], evaluated at the boundaries of the chosen integration domain.

Anonymous on 2023-07-12  [id 3804]

(in reply to Nicolas Wink on 2023-07-10 [id 3792])

The answer of the authors is reasonable and I would suggest to keep this as it is now.

---

## Round 4 · List of Changes

Includes changes suggested by the referee.

- Most notably, updated the convergence figure from some intermediate RG-time to the final RG-time considered throughout the work
- Other changes include typo corrections and extensions to some explanations (see the answer to the referee report for a detailed list)

---

## Editorial Decision

published